# The effect of landfast sea ice buttressing on ice dynamic speedup in the Larsen B embayment, Antarctica

**Trystan Surawy-Stepney**[1], **Anna E. Hogg**[1], **Stephen L. Cornford**[2], **Benjamin J. Wallis**[1], **Benjamin J. Davison**[1], **Heather L. Selley**[1], **Ross A. W. Slater**[1], **Elise K. Lie**[1], **Livia Jakob**[3], **Andrew Ridout**[4], **Noel Gourmelen**[3,5], **Bryony I. D. Freer**[1,6], **Sally F. Wilson**[1], **and Andrew Shepherd**[7]

[1]School of Earth and Environment, University of Leeds, Leeds, United Kingdom

[2]School of Geographical Sciences, University of Bristol, Bristol, United Kingdom

[3]Earthwave Ltd, Edinburgh, Edinburgh, United Kingdom

[4]Department of Earth Sciences, University College London, London, United Kingdom

[5]School of GeoSciences, University of Edinburgh, Edinburgh, United Kingdom

[6]British Antarctic Survey, Cambridge, United Kingdom

[7]Department of Geography and Environmental Sciences, Northumbria University, Newcastle upon Tyne, United Kingdom

**Correspondence:** Trystan Surawy-Stepney (eetss@leeds.ac.uk)

**Abstract.** We observe the evacuation of 11-year-old landfast sea ice in the Larsen B embayment on the East Antarctic Peninsula in January 2022, which was in part triggered by warm atmospheric conditions and strong offshore winds. This evacuation of sea ice was closely followed by major changes in the calving behaviour and dynamics of a subset of the ocean-terminating glaciers in the region. We show using satellite measurements that, following a decade of gradual slow-down, Hektoria, Green, and Crane glaciers sped up by approximately 20 %–50 % between February and the end of 2022, each increasing in speed by more than $100 \, \mathrm{m \, a^{-1}}$. Circumstantially, this is attributable to their transition into tidewater glaciers following the loss of their ice shelves after the landfast sea ice evacuation. However, a question remains as to whether the landfast sea ice could have influenced the dynamics of these glaciers, or the stability of their ice shelves, through a buttressing effect akin to that of confined ice shelves on grounded ice streams. We show, with a series of diagnostic modelling experiments, that direct landfast sea ice buttressing had a negligible impact on the dynamics of the grounded ice streams. Furthermore, we suggest that the loss of landfast sea ice buttressing could have impacted the dynamics of the rheologically weak ice shelves, in turn diminishing their stability over time; however, the accompanying shifts in the distributions of resistive stress within the ice shelves would have been minor. This indicates that this loss of buttressing by landfast sea ice is likely to have been a secondary process in the ice shelf disaggregation compared to, for example, increased ocean swell or the drivers of the initial landfast sea ice disintegration.

## 1 Introduction

The Antarctic ice sheet lost 2671 Gt of ice mass between 1992 and 2020, contributing 7.4 mm towards global sea level rise (Otosaka et al., 2023), with almost all of this loss attributed to ocean-driven ice dynamic processes (Slater et al., 2021). The Antarctic Peninsula (AP) is one of the most rapidly changing parts of the Antarctic ice sheet due to its exposure to increasing air temperatures relative to a warm baseline (Trusel et al., 2015; Banwell et al., 2021), increased ocean forcing (Smith et al., 2020), decrease in ice shelf area (Cook and Vaughan, 2010), and changes in wind and sea ice conditions (Christie et al., 2022; Fraser et al., 2023).

The Larsen B embayment, located on the East AP – bordered by Seal Nunataks in the north and Jason Peninsula in the south – contains 12 major glaciers that flow into the Weddell Sea. Over the last 30 years, satellites have observed major changes in this region. Following a period of rela-

tive stability in the 1990s, the collapse of the Larsen B Ice Shelf in 2002 (Scambos et al., 2004) caused an immediate 8-fold increase in speed (1.0 to $2.8\,\mathrm{km\,a^{-1}}$) on Jorum Glacier, Crane Glacier and the Hektoria–Green–Evans glacier system which were previously buttressed by the ice shelf (Rignot et al., 2004). Though continued buttressing by the Larsen B remnant in Scar Inlet prevented the neighbouring Flask and Leppard glaciers from speeding up in 2002/2003, their ice discharge was 42 % higher by 2013 relative to 1995 (Wuite et al., 2015).

Following the ice shelf collapse in 2002, first-year sea ice formed annually during the austral winter in the Larsen B embayment; however, it remained ice-free each year during the summer months. In 2011, the winter sea ice became land-fast and persisted continuously for 11 years (Christie et al., 2022), before it suddenly disintegrated in January 2022 and was evacuated from the embayment (Ochwat et al., 2023). During the weeks and months following the landfast sea ice evacuation, the floating ice shelves that had developed on the Hektoria–Green–Evans (HGE) glacier system and Crane Glacier (Rott et al., 2018) disintegrated, and their tributary glaciers increased in flow speed (Ochwat et al., 2023). There are clear visual parallels between the clearance of landfast sea ice in 2022 and the disintegration of the Larsen B Ice Shelf 20 years earlier and the subsequent dynamic responses of these glaciers. However, uncertainty remains about the mechanisms driving the increased ice speeds on HGE and Crane in the latter case, particularly regarding the "buttressing" role that the landfast sea ice was able to provide the glaciers prior to collapse (Sun et al., 2023; Ochwat et al., 2023).

Here, we present observations of the spatial pattern of ice speed change before and after the landfast sea ice disintegration in January 2022, along with measurements of glacier calving front location, landfast sea ice extent, and thickness, amongst others. These results complement recent work by Sun et al. (2023) and Ochwat et al. (2023), which document many similar observations to those presented here, though we focus on the glaciers that showed the greatest dynamic response after 2022 – namely Crane and Hektoria–Green–Evans. These observations form the background and motivation for an investigation into the buttressing capacity of the landfast sea ice in which we perform simple diagnostic modelling experiments aiming to quantify the redistribution of stress as landfast sea ice of varying thickness is added to the Larsen B embayment.

## 2 Observations

### 2.1 Observational methods

We collected satellite-derived datasets over the period 2002–2023 to assess changes in landfast sea ice extent, ice shelf extent, and ice flow speed across the Larsen B embayment and the glaciers that terminate there. We used multispectral optical Landsat 8 imagery to visually identify and delineate the landfast sea ice edge in the Larsen B embayment in November each year from 2002 to 2022. We used these Landsat 8 data and additional Sentinel-1 synthetic aperture radar (SAR) backscatter images to manually delineate the calving front location of each glacier feeding the Larsen B embayment (Figs. 1f, S3 in the Supplement). For much of the period prior to the first calving events of 2022 the transition between consolidated ice shelf to landfast sea ice appeared smooth in satellite images, encompassing a region of ice mélange, making the calving fronts difficult to define precisely. However, this does not impact our understanding of the timings of the calving events in 2022. We applied feature tracking techniques to single-look complex Sentinel-1 SAR data collected in Interferometric Wide (IW) swath mode to produce an 8-year record of ice speed over the 12 outlet glaciers in the study region from January 2015 to April 2023 (Davison et al., 2023b). We extracted time series of ice speed averaged over 1 km long flowline segments on 12 glaciers flowing into the Larsen B embayment and used a Kalman smoother with an identity transition matrix to filter the results (Fig. 1a–c) (Wallis et al., 2023). These locations were chosen to be in the centre of the grounded ice streams, close to the grounding line but with enough room for a 1 km buffer along the flowlines that pass through them. Uncertainty indicators in these speeds were calculated by scaling the reciprocal signal-to-noise ratio in the cross-correlation field with the ice speed (Lemos et al., 2018).

### 2.2 Landfast sea ice area change

Our data show that the landfast sea ice went through phases of growth and decay during the period 2002–2011, with sea ice covering the entire embayment at times and only the smaller proglacial embayments at others (Fig. S1). After its formation in winter 2011, the landfast sea ice was retained throughout the summer months until 2022. Its seaward margin retreated and advanced consistently throughout the period 2011–2022, though these oscillations reached a higher amplitude from 2017 onwards (Figs. 1a, S1). Between 18 and 23 January 2022, the multi-year landfast sea ice disintegrated and was evacuated out of the Larsen B embayment (Ochwat et al., 2023), leaving open ocean.

### 2.3 Ice dynamic and calving response

Our velocity measurements show that between October 2014 and January 2022 Hektoria, Green, and Crane glaciers slowed by approximately $100\,\mathrm{m\,a^{-1}}$ (Fig. 1b, d), with smaller decreases in speed on Evans (Fig. 1c, d), Punchbowl, Jorum, and Melville glaciers (Figs. 1b, S2). Over the same period, Flask Glacier sped up slightly by 5 % ($30\,\mathrm{m\,a^{-1}}$), and the remaining four glaciers exhibited fairly stable speeds on annual timescales (Figs. 1c, S2).

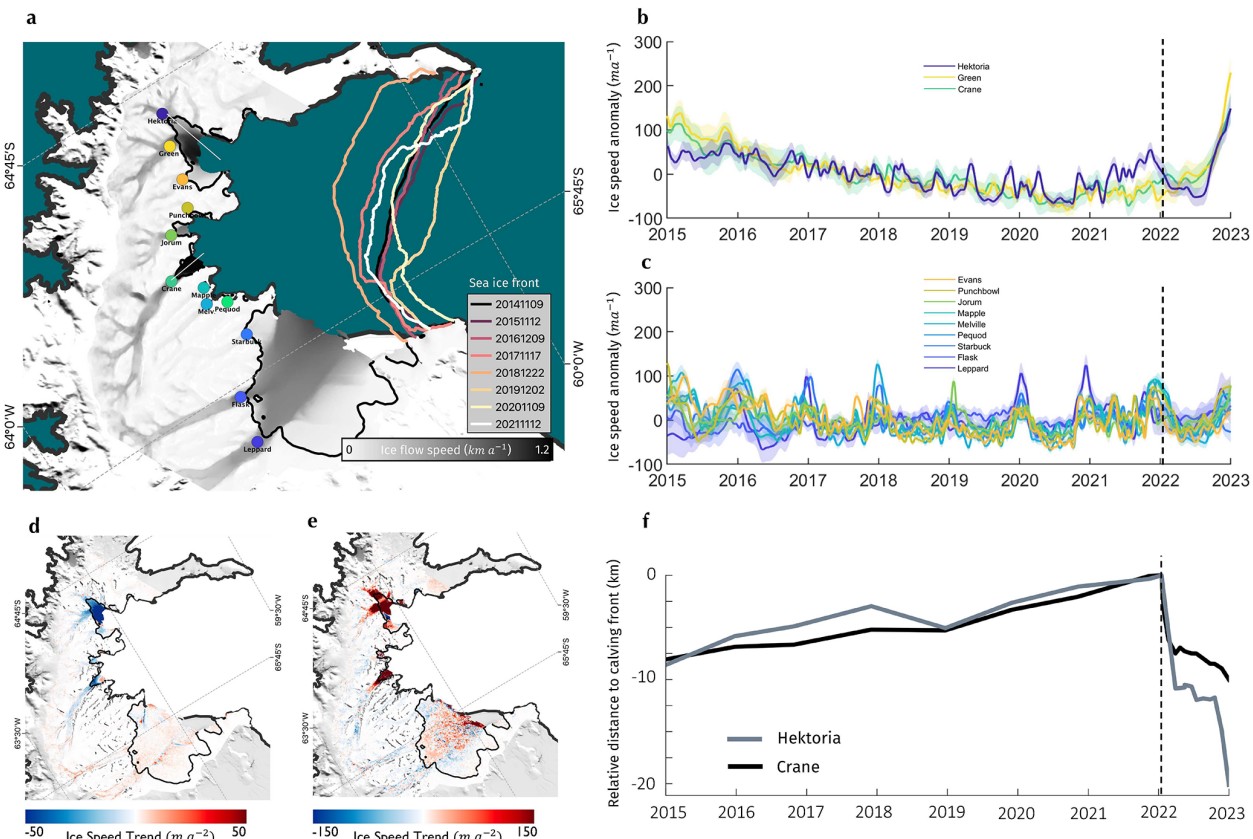

**Figure 1.** Ice speed and speed change map of the Larsen B glaciers. **(a)** Inverse-error-weighted mean ice speed of glaciers flowing into the Larsen B embayment on the East Antarctic Peninsula, measured between October 2014 and April 2023 (greyscale map). Grounding line location (that of Wallis et al. (2024) in HGE and Crane and the grounding line derived by the Making Earth Science Data Records for Use in Research Environments (MEaSUREs) programme's interferometric synthetic aperture radar (InSAR) elsewhere (Mouginot et al., 2017)) are shown with the solid black line. Coloured lines show the landfast sea ice fronts, measured annually between November 2014 and November 2021. **(b)** Ice speed anomaly (signal minus the time series mean) from January 2015 to January 2023 on Hektoria, Green, and Crane glaciers. **(c)** Ice speed anomaly on the other glaciers shown in **(a)**. For the plots in **(b)** and **(c)**, data were extracted over 1 km long segments of flowlines with centre points shown by the circles in **(a)**. The uncertainties shown are $1\sigma$ either side of the mean. **(d)** A map of the observed rate of change in ice speed between October 2015 and October 2021. **(e)** A map of the observed rate of change in ice speed between October 2021 and April 2023, spanning the disintegration event in February 2022. **(f)** Time series of calving front distance on Hektoria Glacier (grey line) and Crane Glacier (black line) from January 2015 to January 2023. These distances are measured along the white lines shown in **(a)**. Data points are annual between 2015 and 2022 and monthly thereafter. Dashed black lines in panels **(b)**, **(c)**, and **(f)** show 18 January 2022, the start of the breakup of the landfast sea ice in the Larsen B embayment. The base map in panels **(a)**, **(d)**, and **(e)** is the Moderate Resolution Imaging Spectroradiometer (MODIS) mosaic of Antarctica (Haran et al., 2021).

During the period of steady or declining speeds between 2014 and 2021 we observe progressive advance of the calving front on the majority of glaciers in the study region. Hektoria and Crane glaciers advanced the furthest, with approximately 12 and 7 km of growth observed between 2015 and 2021 respectively, though this advance was not monotonic (Figs. 1f, S3b). The Larsen B remnant in Scar Inlet is the only calving front to have continually advanced between 2015 and 2021, growing by 6 km. The remaining glaciers experienced changes in calving front position of 3 km or less between 2014 and 2022.

Following the disintegration of the landfast sea ice, we observe a large speedup on Green and Crane glaciers be-

ginning in early to mid-2022 and accelerating from June or July 2022 (Fig. 1b, e), followed by a speedup on Hektoria Glacier beginning in July 2022 (Fig. 1b, e), though this glacier exhibits a more varied signal. At the grounded ice locations chosen for extraction of speed time series, we see changes in speed between January and December 2022 of $35.5 \pm 10.4\%$ on Hektoria Glacier, $46.9 \pm 7.0\%$ on Green Glacier, and $17.8 \pm 5.5\%$ on Crane Glacier. We also see a potential sign change in the ice speed trend on Evans and Jorum (Fig. 1c, e) glaciers in early 2022 from negative to positive, though the changes in speed are comparable to historical variability in the ice speed data. On Hektoria, Green, and Crane glaciers, where the speedup is most pronounced, speed

changes extend up to 10 km upstream of the 2021 grounding line. Our velocity measurements show that there was no pronounced change in speed of Leppard, Flask, Starbuck, Pequod, Melville, Mapple, or Punchbowl glaciers discernible from the variability in ice speed over the preceding decade (Figs. 1c, e, S2).

The large dynamic changes on Crane and Green glaciers were preceded immediately by a period of terminus retreat in February 2022 of 6 and 12 km on the Crane and HGE ice shelves respectively (Fig. 1f). Crane Ice Shelf continued to calve during the period of acceleration and, by the end of our study period, had retreated 12 km relative to its maximum position in December 2022. The HGE Ice Shelf retreated by a further 9 km between September and December 2022, decoupling the ice shelves of Hektoria and Green glaciers (Figs. 1f, S3b).

## 3 Landfast sea ice buttressing

Previous studies focused on the area have shown ice speed changes on HGE and Crane glaciers to be concurrent with changes in terminus position (Wuite et al., 2015; Rott et al., 2018) prior to 2011. In the case of HGE these changes fluctuated, while on Crane steady terminus advance accompanied steadily decreasing glacier speeds. Following the growth of persistent landfast sea ice in 2011, we see persistent terminus advance and decreasing speed, shown here and elsewhere (Wuite et al., 2015; Rott et al., 2018; Ochwat et al., 2023). These observations, along with that of ice shelf disintegration after the sea ice evacuation in 2022 shown here and in Ochwat et al. (2023), suggest a coupling of landfast sea ice to glacier dynamics in which landfast sea ice permitted the growth of the ice shelves in front of HGE and Crane glaciers prior to 2022 which acted as a control on the upstream flow. However, it is unclear the extent to which the landfast sea ice could have itself acted to buttress the upstream glaciers and whether the growth of the ice shelves is attributable to the buttressing effect of landfast sea ice as opposed to other mechanisms by which it can confer stability on regions of ice mélange.

Recent observational reports of the January 2022 evacuation of landfast sea ice from the Larsen B embayment provide conflicting accounts of the possible buttressing effect the landfast sea ice could have had. Ochwat et al. (2023) suggest that the growth of the ice shelves during the residency of the landfast sea ice, the potential dampening of ice speed in Scar Inlet, and the immediate speedup of certain ice shelves following the collapse of the landfast sea ice are evidence of its buttressing effect. However, Sun et al. (2023) suggest that the limited immediate response of the glaciers to the landfast sea ice evacuation and the potential plastic rheological response of the landfast sea ice to sudden changes in upstream stress are reason to believe any buttressing was minimal.

In the context of ice shelves, buttressing refers to the hypothetical difference in englacial stress with and without the ice shelf (Gudmundsson, 2013; Fürst et al., 2016). To be consistent, we take buttressing to have the same meaning in the context of landfast sea ice. Consequently, there are two ways that landfast sea ice buttressing could have contributed to the observed speed changes on the HGE and Crane glaciers by (1) directly influencing the stress distribution in the glaciers such that the disintegration of the landfast sea ice caused an instantaneous speed change on the grounded ice and (2) reducing stresses in the ice shelves which would have otherwise been too great for the ice shelves to withstand. This latter mechanism is a second-order effect of buttressing on grounded ice speed, the implication being that the disintegration of the landfast sea ice in turn caused the disintegration of the ice shelves via loss of buttressing and hence the loss of the ice shelves as a control on the upstream dynamics. This is to be contrasted with other non-buttressing mechanisms by which the landfast sea ice could influence the stability of the ice shelves, such as by bonding fragments of mélange together, preventing small calving events at the glacier terminus, stopping the export of icebergs, and dampening swell-induced loading cycles.

Here, we use the BISICLES ice sheet model (Cornford et al., 2013) to directly investigate these possible effects for the glaciers that exhibited the most pronounced changes in dynamic behaviour after January 2022, namely the HGE system of glaciers and Crane Glacier.

### 3.1 Modelling methods

BISICLES is a finite-volume, adaptive mesh model that solves a discretised form of the shallow-stream approximation to the momentum balance equations,

$$\nabla \cdot [\phi h \bar{\mu}(\nabla \boldsymbol{u} + (\nabla \boldsymbol{u})^{\top} + 2(\nabla \cdot \boldsymbol{u})\mathcal{I})] \\ - Cf(u)\boldsymbol{u} - \rho_{\mathrm{i}}gh\nabla s = 0, \tag{1}$$

where $\boldsymbol{u}$ is the ice velocity, $h$ is the ice thickness, $s$ is the ice surface elevation, $\mathcal{I}$ is the identity operator, $f(u)$ is a function parametrising our sliding law, $C$ is a scalar "basal slipperiness" field, $\rho_{\mathrm{i}}$ is the density of ice, $g$ is the acceleration due to gravity, $\bar{\mu}$ is the vertically averaged effective ice viscosity, and the scalar field $\phi$ is a "stiffness" that scales $\bar{\mu}$. We use Glen's flow law with an exponent of 3, a rate factor according to Cuffey and Paterson (2010), and an internal energy field generated from a continent-wide thermomechanical spin-up in the calculation of the effective ice viscosity. Simulations were carried out at a maximum of 125 or 250 m resolution.

Initially, we set up a model domain with geometry approximately reflecting the HGE and Crane basins of the Larsen B embayment, using a combination of smoothed bedrock elevations according to Huss and Farinotti (2014), grounding line positions time-stamped for the year 2019/20 (Wallis et al., 2024), and surface elevations from the Reference

Elevation Model of Antarctica (REMA) digital elevation model (DEM) (Howat et al., 2019) which is time-stamped to May 2015. Contemporary grounding line locations are required as there has been significant grounding line retreat in recent years (Wallis et al., 2024). The REMA DEM does not reach the edge of the HGE and Crane ice shelves as they were in 2021, so we filled this gap by extrapolating the DEM along flowlines. We performed an inversion for basal slipperiness ($C$) and stiffness ($\phi$) fields using observations of ice speed across the HGE and Crane basins averaged over 2021 (Cornford et al., 2015). At this point we did not include the landfast sea ice in the model geometry, so the glaciers terminate in open sea. The choice to do this presupposes that the inclusion of the landfast sea ice will do little to change the solution to the inverse problem and is necessary as the thickness of the landfast sea ice is not well constrained. We shall see that this assumption is validated by the results. We used regularisation with a Tikhonov operator that approximates the gradients of the control fields to improve the conditioning of the problem. L-curve analysis (Hansen, 1994) was used to select an appropriate level of regularisation (Fig. A1). The gaps in time between the surface elevation, grounding lines, and ice speed data used for the inversions meant that a certain amount of "relaxation" of the geometry was required. We ran five inverse problems separated by a year of thickness evolution. Figure S6 shows geometry and control fields at the end of the model initialisation. We note that, as the thickness and bedrock data in this region of Antarctica are very poorly constrained, the geometry we construct should be considered plausible as opposed to fully accurate. As such, our conclusions are subject to change under replication using different glacier geometries. Given the potential influence of unknown deviations in the real geometry from the data available to us, we avoid performing transient simulations which risk amplifying the impact of these uncertainties which, for the basic mechanical arguments made here, are not likely to be important.

To simulate the effect of the landfast sea ice in the embayment, we assume that it can be treated as a thin ice shelf with the same constitutive ice rheology as the upstream glaciers. Considering a range of length scales and timescales, sea ice is typically treated as a viscoplastic (e.g. Hibler, 1979) or elastoplastic material (e.g. Hunke and Dukowicz, 1997). This captures a material that is strong in compression, weak in extension, and shear, dominated by plastic deformation in thin-ice-covered leads or pressure ridges. Regarding the landfast sea ice that inhabited the Larsen B embayment prior to 2022, satellite images show a fairly uniform, unbroken ice coverage with a smooth deformation field (Fig. S4). This suggests incompressible flow with stress continuity between the landfast sea ice edge and the glacier calving fronts and largely smooth deformation over its decade of residency, though indications of larger-length-scale plastic response to sudden loading have been observed (Sun et al., 2023). Any subcritical viscous deformation internal to the sea ice is controlled by

a rheology which depends on the relative abundances of meteoric and congelation ice, with differences in crystal structure and the presence of brine inclusion leading to a lower effective viscosity in the latter case. Similarly, surface-melt-induced porosity in the meteoric ice would lower its effective viscosity relative to glacier ice. The use of the same formula for the effective viscosity of land and sea ice, along with the assumption of viscous deformation, means that our treatment is to likely provide an upper bound to the buttressing strength that unbroken landfast sea ice could exert on the upstream glaciers. We provide additional evidence in support of this claim in Sect. 5.1.

After setting up the model domain in the way described above, we add such modelled landfast sea ice between the glacier calving fronts and the observed seaward limit of the multi-year landfast ice (Fig. 2a). We then recalculate ice speed over the domain by solving the stress–balance equations in this new configuration. We compare these along flowline transects with the speed in the absence of the landfast sea ice (Fig. 2b–d) and also with flowline transects of quarterly averaged speed observations from before and after the landfast sea ice evacuation. These speed observations are from the second quarter of 2021 (the last before the sea ice evacuation – excluding the months October–December 2021 where high surface melt rendered the data unreliable) and the last quarter of 2022 and are smoothed with a 5 km window. The transects of observations are cut off at the most landward calving front position observed during the quarter or at the edge of our speed observations (which coincide with the edge of the REMA DEM). This gives us the results of Sect. 3.2.1.

For the simulations, we use a regularised Coulomb sliding law (Schoof, 2005; Joughin et al., 2019) with a threshold ice speed of $u_o = 300\,\mathrm{m\,a^{-1}}$ so that sliding is plastic on much of the grounded ice. This ensures that basal stresses remain relatively unchanged as landfast sea ice is introduced, resulting in enhanced changes to the resistive stress and greater ice speed change. Our choice of $u_o = 300\,\mathrm{m\,a^{-1}}$ is based on the speeds of tributary glaciers flowing into HGE and Crane glaciers, though a choice of an even lower threshold makes little difference to the main results (Appendix A3). This was carried out for landfast sea ice thicknesses of 1, 2, and 5 m and from 10 to 50 m in increments of 10 m – a range that extends beyond what one might expect to be realistic landfast sea ice thicknesses in the study region. CryoSat-2 radar altimetry observations show the landfast sea ice had a mean freeboard under 1 m over the period 2013 to 2020 (Fig. S5), implying an approximate thickness under 10 m. We chose to model the landfast sea ice as uniform in thickness, despite these data suggesting otherwise, as the data do not extend to the critical zone at the glacier calving front. This should do little to change the conclusions of the modelling results. The sea ice was assumed to be of vertically and horizontally uniform temperature of $-5\,\mathrm{°C}$. This is likely to lead to stiffer ice with greater buttressing strength than in reality as obser-

vations of surface melt (Ochwat et al., 2023) indicate that the sea ice might be better modelled as temperate.

Additionally, we assess the sensitivity of grounded ice speed to changes in the thickness of the landfast sea ice compared to the thickness of the glaciers themselves (Sect. 3.2.1). This gives us some intuition as to where any changes in geometry that occurred in 2022 might have led to changes in glacier speed. We treat the stiffness $\phi$ as a proxy for ice thickness ($h$) as, to first order, perturbations in these quantities have the same effect on vertically integrated effective viscosity. We consider two 1 km radius circular regions on the grounded ice of Hektoria and Crane glaciers ($\Omega_H$ and $\Omega_C$ respectively) and define the function $\mathcal{J}$ as the mean ice speed over these regions. We then calculate the magnitude of the gradient of $\mathcal{J}$ with respect to $\phi$ using standard adjoint-based methods (Appendix A2). To ensure that we are not aliasing an atypical part of the solution space, we find these sensitivities for six realisations of the control variables $C$ and $\phi$ corresponding to the solutions of the inverse problem with different amounts of regularisation.

Finally, to examine the component of ice shelf stability due to the buttressing of landfast sea ice (Sect. 3.2.2) we further study the case in which 10 m of landfast sea ice is added to the embayment. We look at how strain rates and stresses near the glacier termini change. Principal strain rates $\varepsilon_1$ and stresses $\sigma_1$ (the largest eigenvalues of the strain rate and resistive stress tensors local to each parcel of ice) were calculated and compared for the HGE and Crane ice shelves for 10 m vs. 0 m of landfast sea ice.

## 3.2  Modelling results

### 3.2.1  Direct buttressing of grounded glaciers

Transects located approximately along flowlines of four glaciers that accelerated in 2022 show that speeds along the glaciers change smoothly as a function of landfast sea ice thickness in the embayment (Fig. 2b–e). However, these changes in speed are strongly attenuated upstream of the calving front. At the grounded ice locations used to produce the time series in Fig. 1b, we see instantaneous ice speed changes on the order of 0 %–1 % with the addition of a realistic upper limit of 10 m thick landfast sea ice and a speed change of 0 %–9 % with the highly unlikely thickness of 50 m (Fig. 2b–f). This is considerably below the 15 %–50 % speed change observed on these glaciers (Figs. 1b, 2b–e) after January 2022. These modelled changes in grounded ice speed are to be contrasted with the much larger 2 %–10 % changes in speed seen at the calving fronts and on the floating ice shelves in the simulations with landfast sea ice thickness of 10 m (Fig. 2b–e). These modelled percentage changes in speed are similar in magnitude on all glaciers including Evans, where we do not observe a substantial dynamic response (Fig. 1g). This is an indication in its own right that the buttressing effect of landfast sea ice was not its primary control on the dynamics of the glaciers of the Larsen B embayment.

In the range 0–50 m, the addition of ice in the Larsen B embayment produces changes in speed that vary approximately linearly with thickness (Fig. 2f). We invert this relationship and look, in Fig. 3, at the magnitudes of gradients of ice speed at two locations (Fig. 3a), with respect to effective thickness, assuming 10 m of landfast sea ice. Though Fig. 2 shows that changes in landfast sea ice thickness result in nonzero changes to ice speed along the whole glacier length, we now see that the sensitivity of grounded ice speed to changes in the effective thickness of landfast sea ice is minute in comparison to its sensitivity to changes in the effective thickness of glacier ice. This can be seen especially clearly in transects taken along shear margins of Hektoria and Crane glaciers and out into the landfast sea ice (Fig. 3b.1, b.2). The map of logarithmic sensitivity (Fig. 3a) shows the landfast sea ice in the proglacial embayments to have an impact on the grounded ice, though this diminishes quickly further out to sea. These results further suggest that direct landfast sea ice buttressing of the parts of the glaciers that showed speed increase in 2022 is likely to have been negligible.

### 3.2.2  Buttressing of the floating ice shelves

Despite the fact that speeds in these regions change by a mean of under 4 % except very close to the calving fronts when 10 m of landfast sea ice is added to the embayment (Fig. 2), there are changes in principal strain rates across parts of the HGE and Crane ice shelves on the order of 10 % (Fig. 4a–b), with changes largest in the seaward-most parts of the shear margins. Together with Fig. 2, this indicates that the buttressing effect of 10 m of landfast sea ice in the embayment is enough to produce some dynamic response on the weak floating ice. The important question for the Larsen B glacier system that remains is whether this change could have destabilised the ice shelves of HGE and Crane glaciers.

The simulations exhibit a smooth redistribution of stress when the landfast sea ice is added; however, this is unrealistic for highly crevassed ice shelves where thicknesses can vary considerably from place to place. Continuity instead ensures that the vertically integrated stress is concentrated in thinner areas. It is possible that small changes in this stress would surpass the load-bearing capacity of these thinner sections of glacier ice, causing the ice shelf to break up. To assess how plausible this is, we compare the distributions of vertically averaged principal resistive stresses across the ice shelves with and without 10 m thick landfast sea ice (Fig. 4c–d). We see on both HGE and Crane glaciers that the addition of the landfast sea ice reduces the mean principal stress; however, this shift is small compared to the variance of stresses within the ice shelves. Assuming that weaknesses in the ice shelves are not concentrated in a small region, it is improbable, therefore, that the ice shelves were stable prior to the landfast sea ice removal and that this removal caused a large

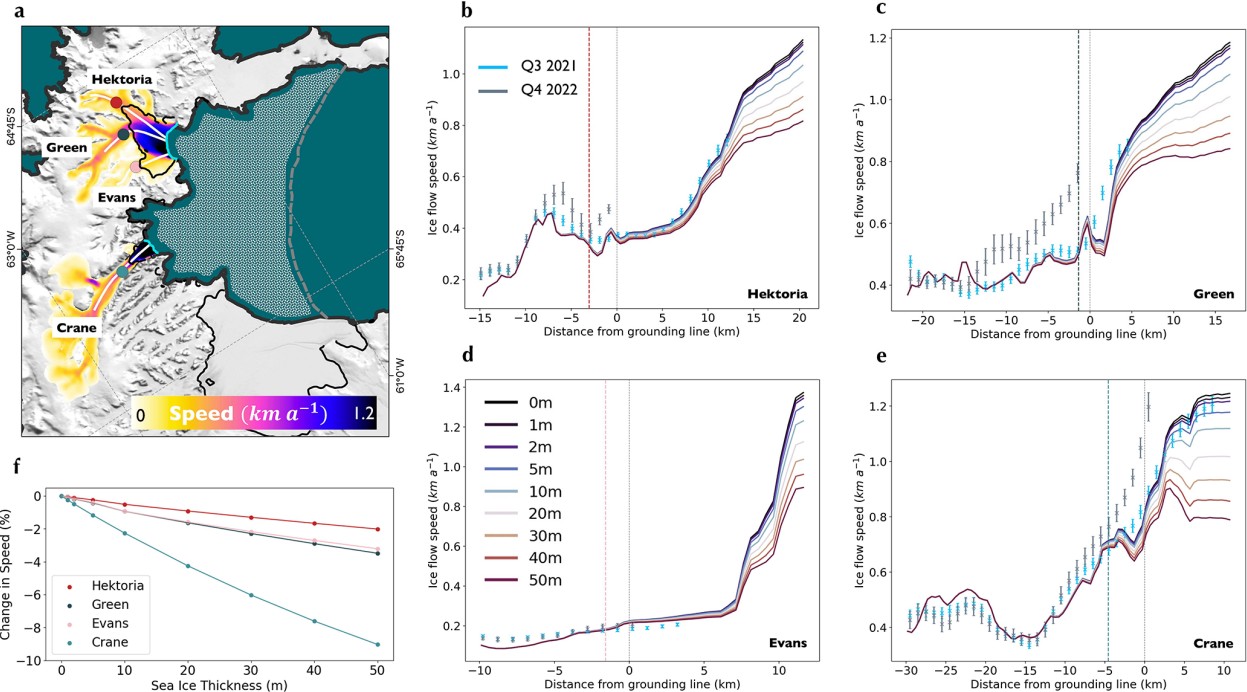

**Figure 2.** Modelled changes in speed with varying landfast sea ice thickness. **(a)** The Larsen B embayment. Flow speeds following the inversions for stiffness and basal slip coefficient fields over the Hektoria and Crane basins. The patterned area bound by the embayment walls and the dashed grey line indicate where landfast sea ice was added during the simulations. Coloured circles on the glaciers show where time series of speed were extracted in Fig. 1. Flowlines used to extract speeds for different landfast sea ice thicknesses in **(b)**–**(e)** are shown in white. The base map is the MODIS mosaic of Antarctica (Haran et al., 2021). **(b–e)** Modelled ice speeds for different landfast sea ice thicknesses, where dark blue indicates 0 m and dark red indicates 50 m, along the flowlines shown in **(a)** for **(b)** Hektoria, **(c)** Green, **(d)** Evans, and **(e)** Crane. Thin vertical dashed grey lines show the positions of the grounding lines, and coloured vertical lines show the positions of the corresponding circles in **(a)**. Points plotted in cyan show the ice speed measured along these transects in the third quarter of 2021 – smoothed with a 5 km window. Points plotted in grey show the equivalent for the last quarter of 2022. **(f)** Percentage change in modelled ice speed for different landfast sea ice thicknesses at the locations with colours is shown in **(a)**.

enough perturbation in resistive stress to account for the spatially extensive ice shelf breakup that was observed.

The above analysis relates to instantaneous changes in the stresses and strain rates within the floating ice that result
5 from a loss of landfast sea ice buttressing. We have suggested that the distributions of principal resistive stresses within the floating ice change, but by an amount that is unlikely to have led to the rapid collapse of the ice shelves such as seen on the Crane and HGE glaciers. However, changes in strain rate
10 (Fig. 4a–b) can have implications for ice shelf stability on longer timescales. For example, increased ice shelf thinning rates, resulting from enhanced velocity gradients, can lessen its robustness to fracturing. Additionally, elevated strain rates can lead to faster changes to the glacier geometry. We cannot
15 rule out the hypothesis that such processes were in part responsible for the elevated calving rate on Crane and HGE ice shelves starting in September 2022. However, if this were the case, it seems likely that the calving events themselves would have had a greater impact on subsequent calving rate than the
20 loss of sea ice.

## 4  Environmental drivers

The aim of this article is to address the potential buttressing capacity of landfast sea ice in light of its evident relationship to glacier dynamics shown by recent events in the Larsen B embayment. However, we briefly diverge here to 25 note a couple of interesting climatic factors that may have played a part in the 2022 landfast sea ice fragmentation and the subsequent dynamic response of the glaciers. To examine these, we looked at air temperature and wind velocity data from ERA5 reanalysis over the Larsen B embayment 30 between 2000 and 2022 and estimates of basal melt rate from swath mode CryoSat-2 radar altimetry data acquired between November 2010 and January 2022 (Gourmelen et al., 2017; Davison et al., 2023a).

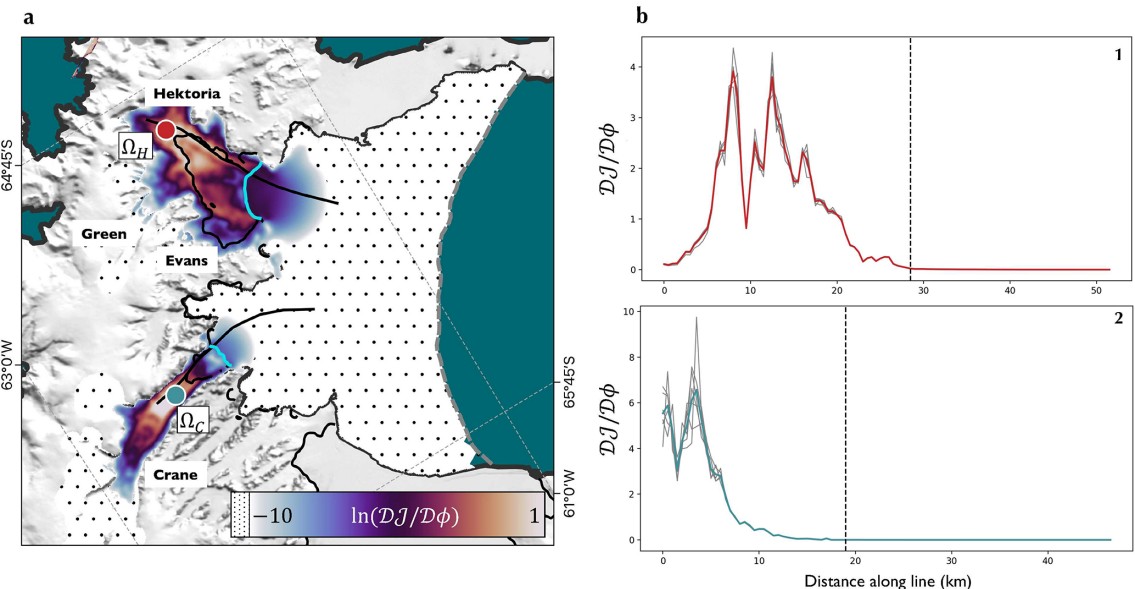

**Figure 3.** Sensitivity of ice speed to changes in effective ice thickness. **(a)** Magnitudes of differential sensitivities of ice speed in the locations marked by the coloured circles to change in stiffness across the domain. The spotted black area shows where the sensitivity is under $e^{-10}$. Cyan lines show the boundaries between the glacier termini and the start of the landfast sea ice according to the model geometry. Black lines show where transects of modelled speeds were collected to produce the graphs in **(b)**–**(c)**. The base map is the MODIS mosaic of Antarctica (Haran et al., 2021). **(b)** Magnitudes of sensitivities along the transects shown in **(a)**. Dashed black lines show the location along the transect of the glacier terminus for **(b.1)** Hektoria and **(b.2)** Crane. Grey lines indicate the different realisations of the control variables $C$ and $\phi$, while coloured lines show the mean sensitivities. Note that data are presented logarithmically in **(a)** and linearly in **(b)**.

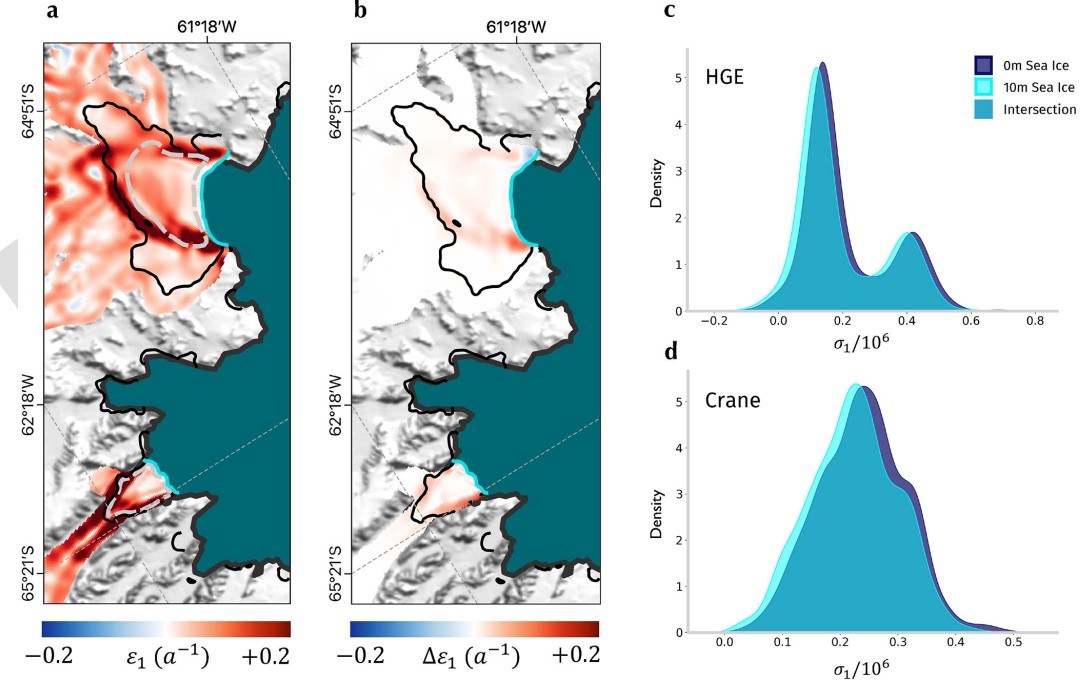

**Figure 4.** Change in the modelled strain rates and stresses in response to the addition of 10 m thick landfast sea ice. **(a)** Principal strain rate across the HGE and Crane ice shelves with 10 m thick landfast sea ice. **(b)** Difference in principal strain rate across the HGE and Crane ice shelves with no sea ice compared to 10 m. The base map in **(a)**–**(b)** is the MODIS mosaic of Antarctica (Haran et al., 2021). **(c–d)** Histograms of principal stress for 10 m thick landfast sea ice (cyan) compared with 0 m (navy) and their overlap over the HGE and Crane ice shelves respectively. These graphs are produced for the floating ice regions bounded with the dashed grey lines in **(a)**. CE2

ERA5 reanalysis data suggest that annual mean surface air temperatures over the 11 years of landfast sea ice residence had been steadily increasing at a rate of $0.25\,°C\,a^{-1}$ over the Vaughan and Exasperation inlets (Fig. 5b). By 2022, the air temperature had increased to $2\,°C$ above the 2000–2022 mean in the Larsen B embayment. ERA5 temperature data show an even more pronounced localised peak in the air temperature anomaly over the Larsen B embayment in the months prior to the landfast sea ice disintegration. These data suggest the possibility that the landfast sea ice in the Larsen B embayment was unable to persist through a longer and more intense melt season than it had encountered in previous years, brought about by trends in atmospheric conditions. However, work by Ochwat et al. (2023) using estimates of melt from passive microwave data shows a peak in melt days in the summer of 2019/20. Additionally, ERA5 wind velocity data suggest there were anomalously strong north-westerlies over the Antarctic Peninsula in 2022 compared to the 2000–2022 mean (Fig. 5c). These strong offshore winds could well have contributed to the landfast sea ice disintegration and would have aided its evacuation from the embayment before it could refreeze as pack ice.

Estimates of ice shelf basal melt rates over Scar Inlet, the last remaining remnant of the original Larsen B Ice Shelf, show that the highest melt rates are located at the grounding line (Fig. 5d). A time series of the mean basal melt rate from this region shows that the rates were fairly constant at $0\pm4\,m\,a^{-1}$ for the majority of the period from 2010 to 2018, after which basal melt rates generally increased, up to a maximum of $6\pm4\,m\,a^{-1}$ in January 2023 (Fig. 5e). While there is an absence of direct ocean temperature measurements during the period around 2021, it is possible that the strong winds shown by the ERA5 data (Fig. 5a) drove an upwelling in ocean circulation bringing warm water up from depth, in addition to blowing out the disintegrated sea ice. If this proxy is representative of changes in ocean temperature across the Larsen B embayment, then it indicates that increased grounding line ablation could have had a role to play in the ice dynamic changes seen on HGE and Crane glaciers. Further work is required to establish whether the deeply grounded glaciers of the Larsen B embayment, and perhaps beyond, exhibited a dynamic signal before the landfast sea ice evacuation that could point to the influence of enhanced sub-shelf or grounding line ablation. While the basal melt rates observed on Scar Inlet since 2020 of up to $10\,m\,a^{-1}$ (Fig. 5d) have not yet caused a notable speedup on Flask and Leppard glaciers, which may be in part explained because they remain buttressed by the laterally constrained ice shelf remnant, these basal melt rates are comparable to those observed on ice streams flowing into the Amundsen Sea sector of West Antarctica (Shean et al., 2019). If sustained or even increased in the future, these basal melt rates may suggest that a dynamic response on these glaciers could be expected in the longer term.

## 5 Discussion

Given the abundance of landfast sea ice fringing the fast-flowing margin of the Antarctic ice sheet (Fraser et al., 2021) and the potentially large contribution of ice dynamics to future changes to Antarctic mass balance (Joughin and Alley, 2011; Pattyn and Morlighem, 2020), understanding how changes in landfast sea ice extent and thickness alters the dynamic behaviour of glaciers is evidently valuable. The sudden evacuation of the landfast sea ice from the Larsen B embayment and the changes to the dynamics of HGE and Crane glaciers that followed have provided us with a natural opportunity to investigate these relationships. The concurrency of the evacuation of the landfast sea ice and the observed changes in calving behaviour and dynamics of the upstream glaciers demonstrates the crucial impact of landfast sea ice in the region; however, our modelling results suggest that the component of this due to buttressing, as it is understood in the context of ice shelves, is likely to have been minimal.

### 5.1 The use of a viscous flow model

The conclusions of this article rely on the assertion that model we chose to use, which treats landfast sea ice in the same way as the land ice, gives an upper bound on the buttressing that the landfast sea ice could provide. We discuss in Sect. 3.1 some of the reasons why this is likely to be true. Here, we provide additional evidence for this by comparing stresses in the modelled landfast sea ice to plausible yield stresses in a widely used viscoplastic sea ice model, namely that of Hibler (1979). In general, sea ice rheology is modelled as being plastic in the case of large deformation due to the opening of cracks, raising of pressure ridges, and shearing along crack boundaries. In the case of small-scale deformation, the rheology is sometimes argued to be elastic, reflecting the interaction of floes as they bump into each other (Coon et al., 1974), or viscous, as an approximation to the random jostling of small floes together (Hibler, 1977). The continuity of speed across the glacier–landfast sea ice boundary, as well as the smooth deformation field we observe (Fig. S4), indicates that the deformation of the landfast sea ice in the Larsen B embayment, if not dominated by internal viscous deformation, is unlikely to have been in such a subcritical regime. The model of Hibler (1979) uses a rheology of the form

$$\sigma_{ij} = 2\eta\dot{\varepsilon}_{ij} + \left[(\zeta - \eta)\dot{\varepsilon}_{kk} - \frac{P}{2}\right]\delta_{ij}, \tag{2}$$

where $P$ parametrises the strength of the ice, and $\zeta \equiv \zeta(\dot{\varepsilon}_{ij}, P)$ and $\eta \equiv \eta(\dot{\varepsilon}_{ij}, P)$ are bulk and shear viscosities respectively. These are decreasing functions of strain rate invariants such that the stress states for typical strain rates lie on an elliptical yield curve. We define $\sigma_I = \frac{1}{2}(\sigma_1 + \sigma_2)$ and $\sigma_{II} = \frac{1}{2}(\sigma_2 - \sigma_1)$, where $\sigma_1$ and $\sigma_2$ are the principal stresses. Assuming isotropic ice, the yield curve then can be plotted as

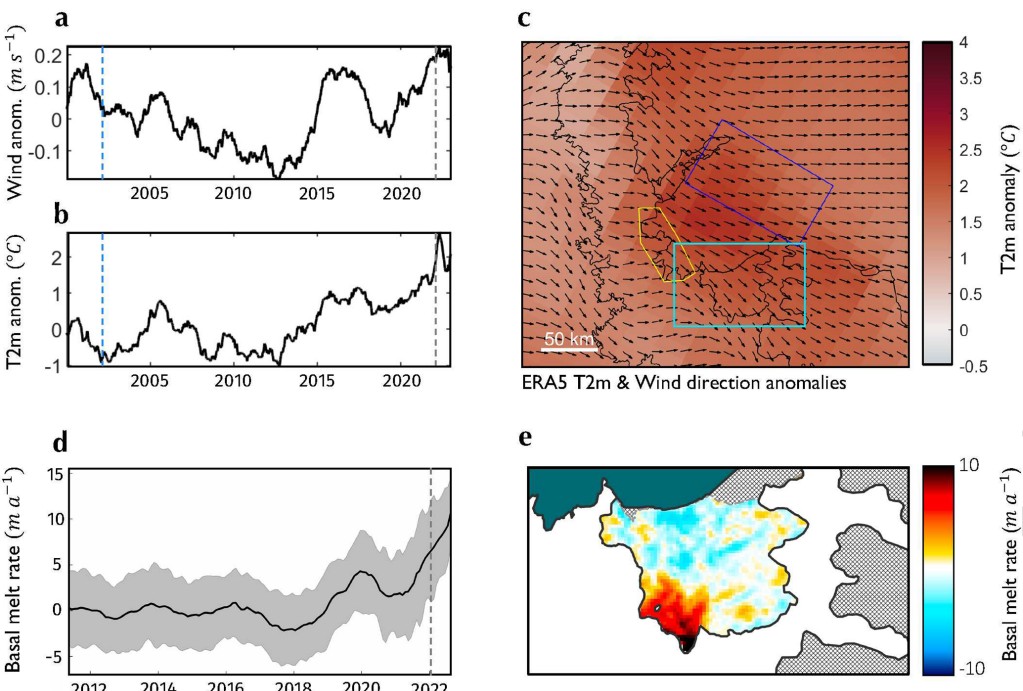

**Figure 5.** Environmental forcing over the Larsen B embayment. **(a–c)** Wind velocity and air temperature data over the period 2000–2022. **(a)** 2022 wind speed anomaly (signal compared to time series mean) extracted from the purple box shown in **(c)**. **(b)** 2 m air temperature anomaly extracted from the yellow box shown in **(c)**. **(c)** Normalised wind direction anomaly (vector field) and 2 m air temperature anomaly (colour) over the Larsen B embayment in 2022 compared to the 2000–2022 mean. Vertical dashed blue lines indicate March 2002 – when the Larsen B Ice Shelf disintegrated – and vertical dashed grey lines show January 2022 – when the landfast sea ice disintegrated. **(d–e)** CryoSat-2 swath mode ice shelf basal melt rate observations from November 2010 to January 2022 over Scar Inlet in the Larsen B embayment. **(d)** Time series of mean monthly basal melt rate in Scar Inlet (indicated with a blue box in **c**). The vertical dashed grey line shows January 2022. **(e)** Mean basal melt rate over the Scar Inlet (shown by the blue box in **c**) between November 2010 and January 2022.

a function of $\sigma_{\mathrm{I}}$ and $\sigma_{\mathrm{II}}$ (Feltham, 2008). This curve passes through the origin, has major axis width of $P$ and centre at $(-P/2, 0)$, and has an eccentricity that depends on the relative strength of the ice in shear and compression. The ice strength itself is approximated as follows:

$$P = P^* h A e^{-c^*(1-A)}, \tag{3}$$

where $P^* = 2.75 \times 10^4 \, \mathrm{N\,m^{-2}}$, $c^* = 20$, $h$ is the sea ice thickness, and $A$ is its concentration. $P^*$ is sometimes treated as a tunable parameter, but $P$ is greater than a factor of 10 away from $10^5 \, \mathrm{N\,m^{-1}}$ CE3 (Feltham, 2008). Regardless of the precise subcritical rheology, the strength parameter $P$ is the key scale for stresses that can be maintained within the landfast sea ice.

We consider a yield curve for the landfast sea ice in the Larsen B embayment with $P = 10^6 \, \mathrm{N\,m^{-1}}$ and an eccentricity of 0. We consider this to be a "maximal" yield curve as $P$ is likely to be smaller in reality, and sea ice is generally far weaker in shear than in compression. (We also plot a more realistic yield curve with an eccentricity of 2 and $P = 2.75 \times 10^5 \, \mathrm{N\,m^{-1}}$ as suggested in Hibler, 1979.) We compare the resistive stresses in the modelled landfast sea ice to this yield curve, paying special attention to the

proglacial embayments in front of the HGE and Crane ice shelves (Fig. 6). The resistive stresses largely lie far outside the yield curve and those in the proglacial embayments all do. This indicates that the stresses born by the landfast sea ice in our model are substantially larger than would be expected of real landfast sea ice. Hence, our modelled landfast sea ice has a considerably greater buttressing capacity than could be expected from a more realistic and well-known model.

## 5.2 What impact does sea ice have on glacier ice?

We have seen that, given a definition of buttressing analogous to that of ice shelves, landfast sea ice has limited ability to buttress glaciers, essentially due to its relative thinness. However, there is a clear link between the landfast sea ice and the stability of the glaciers in the region. As mentioned in Sect. 3, one way in which this could occur is through the promotion of ice shelf growth, for example, by the following mechanism. Landfast sea ice allows ice shelves to grow initially by providing a barrier to icebergs that, having calved from the glacier terminus, would otherwise drift away. Sea ice formation between these icebergs bonds them together and to the glacier calving font, forming a rigid ice mélange

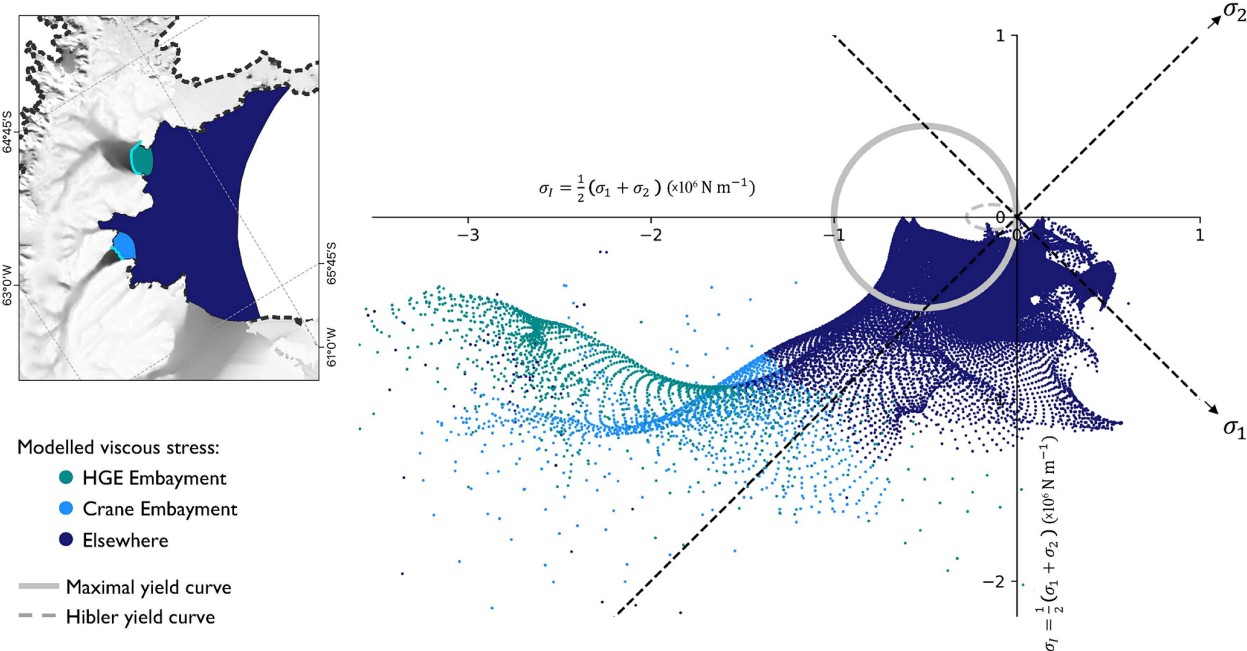

**Figure 6.** A comparison of resistive stresses in the modelled landfast sea ice and a possible yield curve for sea ice. The graph shows resistive stresses in the modelled landfast sea ice plotted as a function of "negative pressure" $\sigma_I$ and "maximum shear stress" $\sigma_{II}$ (Feltham, 2008). The colour of the points indicates where in the domain the modelled resistive stress is extracted, corresponding to regions on the inset map. The grey circle on the graph indicates a yield curve for a possible sea ice rheology of the form of Eq. (3) with $P = 10^6 \, \mathrm{N \, m^{-1}}$ and an eccentricity of 0. The dashed grey ellipse shows a yield curve with an eccentricity of 2 and $P = 2.75 \times 10^5 \, \mathrm{N \, m^{-1}}$ as suggested in Hibler (1979). The base map in the inset map is the MODIS Mosaic of Antarctica (Haran et al., 2021).

with material properties determined by the size and density of the icebergs and the strength of the sea ice bonds. These proto-ice shelves get progressively more robust over time as icebergs calving upstream are more tightly constrained and sea ice thickens. Such a mechanism would lead to ice shelves with no clear boundary in thickness or rigidity separating them from the landfast sea ice – such as those in the HGE and Crane glaciers prior to 2022 – but rather a gradient in those properties. Hence, landfast sea ice confined to an embayment could stabilise ice shelves through the twin mechanisms of inhibiting the export of icebergs and increasing the strength of bonds between them. If this landfast sea ice disintegrates, the visible calving of mélange at the seaward end of the ice shelf would increase through the first mechanism and, if ocean or atmospheric conditions led to the breakup of sea ice within the ice shelf itself, some level of fragmentation of the mélange would also be expected. This is perhaps the process responsible for the partial fragmentation of the Crane and HGE ice shelves concurrently with the landfast sea ice evacuation, which led to the speedup of Crane and Green glaciers.

Perhaps more importantly for the additional calving events seen on the Crane Ice Shelf in the months following the landfast sea ice evacuation and those on Hektoria and Green glaciers in September 2022, sea ice can act to attenuate ocean swell that originates outside of the embayment (e.g. Voer-

mans et al., 2021). This occurs through scattering of ocean waves from heterogeneities in the ice cover and the dissipation of wave energy, for example, by the elastic plate bending of the sea ice at different length scales and the short wavelength fracturing of ice near its margin (Squire, 2020). Without the sea ice, ocean swells cause higher-amplitude flexural loading cycles that can lead to supercritical and subcritical brittle failure (Holdsworth and Glynn, 1978; Massom et al., 2018). The observations of immediate fragmentation of weak seaward parts of the HGE and Crane ice shelves in January 2022 and the continuation of enhanced calving on the HGE and Crane ice shelves months after the evacuation of the landfast sea ice are compatible with the idea that the ice shelves were not capable of withstanding persistent elevated swell after the breakup or even large swell events leading to critical failure such as that proposed by Ochwat et al. (2023) as a cause of the landfast sea ice disintegration itself.

### 5.3 Wider implications for the interaction of glaciers with landfast sea ice

The results presented here are relevant to a certain limit of landfast sea ice and glacier conditions. (i) The landfast sea ice existed in a relatively enclosed embayment. (ii) It was likely to have been relatively thick due to its multi-year persistence over the previous 11 years in a relatively cold ocean. (iii) It

appeared spatially coherent and deformed smoothly. (iv) The inflowing ice shelves were rheologically weak. How well the specific results generalise depends on how well these variables apply to the situation under consideration. However, the conclusion that unbroken landfast sea ice has limited capacity to directly buttress glaciers is likely to hold in general as the conditions explored here are consistent with a "maximum buttressing" example. (The exception is in the rheology of the ice shelves, where stiffer ice may be more capable of transmitting stress upstream.)

In recent decades, the general picture of Antarctic sea ice has been one of regional fluctuation and relative continental stability, with sea ice extent (SIE) increasing slightly between the early 1970s and the mid-2010s (Turner et al., 2022). The year 2014 saw the beginning of a fall in SIE, culminating in successive years of record Antarctic sea ice lows in 2022 and 2023 (Turner et al., 2022; Purich and Doddridge, 2023).

Several studies have indicated the importance of sea ice in maintaining glacier stability in Antarctica and Greenland (Arthur et al., 2021; Christoffersen et al., 2012) and the results of this study do little to suggest otherwise, merely highlighting that this importance is unlikely to stem from its ability to buttress glaciers in the way an ice shelf can buttress a grounded ice stream. However, to more accurately judge the extent to which landfast sea ice stabilises ice shelves, a greater understanding of the mechanisms by which this happens is required. To help with observational and modelling studies that aim to do this, work should be carried out to close the gaps in landfast sea ice extent, concentration, and altimetry measurements at the critical zone near glacier calving fronts. For example, knowledge of sea ice thickness at the point of contact with the calving front, along with calving front morphology, would help better our understanding of processes in which sea ice might inhibit iceberg calving via its influence on torques at the glacier front.

## 5.4 Limitations and future work

The geometry used in our modelling experiments was one in which glacier ice gave way suddenly to sea ice. In reality, in early January 2022, the boundaries between the HGE and Crane ice shelves and the landfast sea ice in front of them were less obvious. The transition between glacier and sea ice involved stages of increasingly rarefied mélange, including icebergs of a range of sizes. It is quite possible that this transition zone has the dual properties of being able to supply meaningful buttressing to the upstream ice shelf through interactions between icebergs and of being itself vulnerable to the loss of landfast sea ice. Idealised configurations not dissimilar to this were investigated in Robel (2017). It is very possible that the dynamics of these regions played a role in the destabilisation of the parts of the HGE and Crane ice shelves that disintegrated immediately after the landfast sea ice evacuation. That these regions of mélange disaggregated

at the time of the sea ice disintegration could suggest a buttressing of them by the landfast sea ice and hence a second-order buttressing effect of the landfast sea ice on the glaciers. However, it seems more likely, though more mundane, that the more rarefied parts of the mélange would be susceptible to the same forcing as the landfast sea ice and so disintegrated in January 2022 for the same reasons. Future work should look in greater detail at the mechanisms by which landfast sea ice can interact with glaciers through such transitional zones of ice mélange, as these might be key to the coupling in embayment and fjord-like geometries.

The sensitivities presented in Fig. 3 are reported for a number of realisations of the control fields $C$ and $\phi$. This is because, by looking at solutions corresponding to different amounts of regularisation, we hope to show that the spatial pattern of ice speed sensitivities is typical for solutions near the misfit minimum. A more complete picture of the sensitivities might be obtained in future by looking at the curvature of ice speed around the solution, i.e. the principal components of the Hessian matrix. At present, such analysis is difficult to achieve in BISICLES but is possible for models employing automatic differentiation (e.g. Recinos et al., 2023). Additionally, this would enable a more exact computation of the gradient, rather than the linear approximations used here (Goldberg and Sergienko, 2011).

Additional observations will help to further our understanding of the relative importance of the components of the effect of landfast sea ice on the stability of floating glacier ice. We see in the sensitivity maps (Fig. 3) a focusing of the effect of landfast sea ice thickness close to the glacier termini. This suggests that the buttressing component of the effect of landfast sea ice on the ice shelves is concentrated close to the glaciers. However, the attenuation of ocean swell might rely on the full extent of sea ice in the embayment and beyond (Ochwat et al., 2023). Future observations of sea ice growth in the small embayments, e.g. seasonally, vs. the Larsen B embayment as a whole and the impacts on the calving behaviour of Hektoria and Crane glaciers can help shed more light on the relative importance of these processes on the growth of ice shelves.

## 6   Conclusions

Our results show that multi-year landfast sea ice which had been present in the Larsen B embayment for the last 11 years, following the collapse of the Larsen B Ice Shelf in 2002, completely disintegrated between 18 and 23 January 2022. This was followed in February by the onset of major ice dynamic speedup events and changes in the calving behaviour of glaciers flowing into the Larsen B embayment. Hektoria, Green, and Crane glaciers sped up by approximately 15 %–50 % between February and December 2022, with the most pronounced increase of approximately $240\,\mathrm{m\,a^{-1}}$ on Green Glacier upstream of the grounding line. These glaciers lost

the majority of their floating ice shelves, which had built up over the preceding decade, by the end of 2022, with the largest retreats of 12 and 6 km on HGE and Crane glaciers immediately following the loss of the landfast sea ice.

Model simulations suggest that the increases in speed on the now tidewater parts of Hektoria, Green, and Crane glaciers are not due to the loss of direct mechanical buttressing supplied by the landfast sea ice that formerly covered the Larsen B embayment. However, the landfast sea ice undoubtedly had an effect on the floating parts of these glacier systems. This effect can be partitioned into the bonding of mélange by sea ice in the ice shelves, the dampening of ocean swell that would otherwise cause high-amplitude stress cycles in the ice shelves, and the buttressing that reduces internal stresses. The modelling and observations presented here suggest that direct buttressing of the landfast sea ice could have been large enough to have had a dynamic impact on the floating ice but that the disintegration of the ice shelves is unlikely to have been related to the associated small changes in resistive stress. This leads us to suggest that the term buttressing should not be used in the context of sea ice in the way it is understood when applied to ice shelves. However, a more complete model of the glaciers, ice shelves, and landfast sea ice (including the transition zone of ice mélange) is required to fully quantify the relative importance of the effects of sea ice on floating glacier termini, along with further observations of sea ice–ice shelf interactions in the Larsen B embayment and elsewhere.

## Appendix A: Additional model details

### A1    L-curve analysis

As discussed briefly in Sect. 3.1, we use a form of Tikhonov regularisation to replace the ill-posed inverse problem with a "nearby" well-posed one, with an operator that calculates spatial gradients of $C$ and $\phi$. The inverse problem can be written as

$$\underset{C,\phi}{\mathrm{argmin}}\left\{\int_\Omega |u-u_\mathrm{o}|^2\,\mathrm{d}\Omega + \int_\Omega \left(\alpha_\phi|\nabla\phi|^2 + \alpha_C|\nabla C|^2\right)\mathrm{d}\Omega\right\},$$
$$\text{s.t.}\quad G(u,C,\phi)=0, \tag{A1}$$

where $u$ is the modelled ice speed, $u_\mathrm{o}$ is the observed ice speed, and $G(u,C,\phi)=0$ is the shallow-stream momentum balance Eq. (1). This is approximately solved in BISICLES using a non-linear conjugate gradient method.

We use L-curve analysis to find optimal values of $\alpha_C$ and $\alpha_\phi$ (Fig. A1). This is a heuristic method that posits that the optimal values of the regularisation parameters lead to a solution that balances sensitivities of the misfit and regularisation parts of the cost function to changes in their relative weights (Hansen, 1994). Given the tendency for L-curve analysis to over-regularise, we take the values to the immediate right of

the position of maximum curvature in the L-curves, namely $\alpha_C = 1$ and $\alpha_\phi = 10^9$.

### A2    Calculating gradients of ice speed using the model adjoint

Figure 3 displays the gradient of the functional as follows:

$$J(u(C,\phi)) = \int |u|\,\mathrm{d}\Omega_\mathrm{HC}$$

with respect to the field $\phi$. Here, the domain $\Omega_\mathrm{HC}$ is a union between the neighbourhoods marked by coloured circles ($\Omega_\mathrm{H}$ and $\Omega_\mathrm{C}$) in Fig. 3a. We write $J(u(C,\phi))$ as $\tilde{J}(C,\phi)$, a pure functional of $C$ and $\phi$. The Gâteaux derivative of $\tilde{J}(C,\phi)$ with respect to $\phi$ in the direction $\delta\phi$ can be written as

$$\langle D\tilde{J}, \delta\phi\rangle = \lim_{\epsilon\to 0}\frac{\tilde{J}(C,\phi+\epsilon\delta\phi) - \tilde{J}(C,\phi)}{\epsilon}.$$

We approximate this as

$$\langle D\tilde{J}, \delta\phi\rangle \approx \int \delta\phi$$
$$\times \bar{\mu}h\nabla\boldsymbol{\lambda}\left(\nabla\boldsymbol{u} + (\nabla\boldsymbol{u})^\top + 2(\nabla\cdot\boldsymbol{u})\mathcal{I}\right)\mathrm{d}\Omega, \tag{A2}$$

where $\boldsymbol{\lambda}$ is a vector field of Lagrange multipliers that solves the adjoint equation

$$-\nabla\cdot[\phi h\bar{\mu}(\nabla\boldsymbol{\lambda} + (\nabla\boldsymbol{\lambda})^\top + 2(\nabla\cdot\boldsymbol{\lambda})\mathcal{I})] + C\boldsymbol{\lambda}$$
$$= \begin{cases} \hat{\boldsymbol{u}}, & \text{in } \Omega_\mathrm{HC} \\ 0, & \text{elsewhere} \end{cases} \tag{A3}$$

with reflection boundary conditions on the domain boundary

$$\hat{\boldsymbol{n}}\cdot\boldsymbol{\lambda} = 0,$$
$$\hat{\boldsymbol{t}}\cdot\nabla\boldsymbol{\lambda}\cdot\hat{\boldsymbol{n}} = 0$$

(where $\hat{\boldsymbol{n}}$ and $\hat{\boldsymbol{t}}$ are normal and tangent vectors to the boundary respectively).

A more detailed exposition of this kind of procedure is given in Morlighem et al. (2013). To construct Eq. (A3), we have neglected non-linearities in the dependence of $\bar{\mu}$ on $\boldsymbol{u}$ and in the sliding law (Goldberg and Sergienko, 2011). The field we show in Fig. 3 comes from interpreting Eq. (A2) as the projection of the functional gradient along the direction $\delta\phi$ with the standard $L_2$ inner product. Hence, the gradient shown in Fig. 3 is the field

$$\bar{\mu}h\nabla\boldsymbol{\lambda}\left(\nabla\boldsymbol{u} + (\nabla\boldsymbol{u})^\top + 2(\nabla\cdot\boldsymbol{u})\mathcal{I}\right).$$

### A3    Sensitivity to sliding physics

For the simulations presented in this article, we used a regularised Coulomb sliding law for basal stress $\boldsymbol{\tau}_\mathrm{b}$ in terms of basal ice velocity $\boldsymbol{u}_\mathrm{b}$ of the form

$$\boldsymbol{\tau}_\mathrm{b} = -\left(\frac{|\boldsymbol{u}|_\mathrm{b}}{|\boldsymbol{u}_\mathrm{b}|/u_\mathrm{o}+1}\right)^m\frac{\boldsymbol{u}_\mathrm{b}}{|\boldsymbol{u}_\mathrm{b}|}\ \text{TS1} \tag{A4}$$

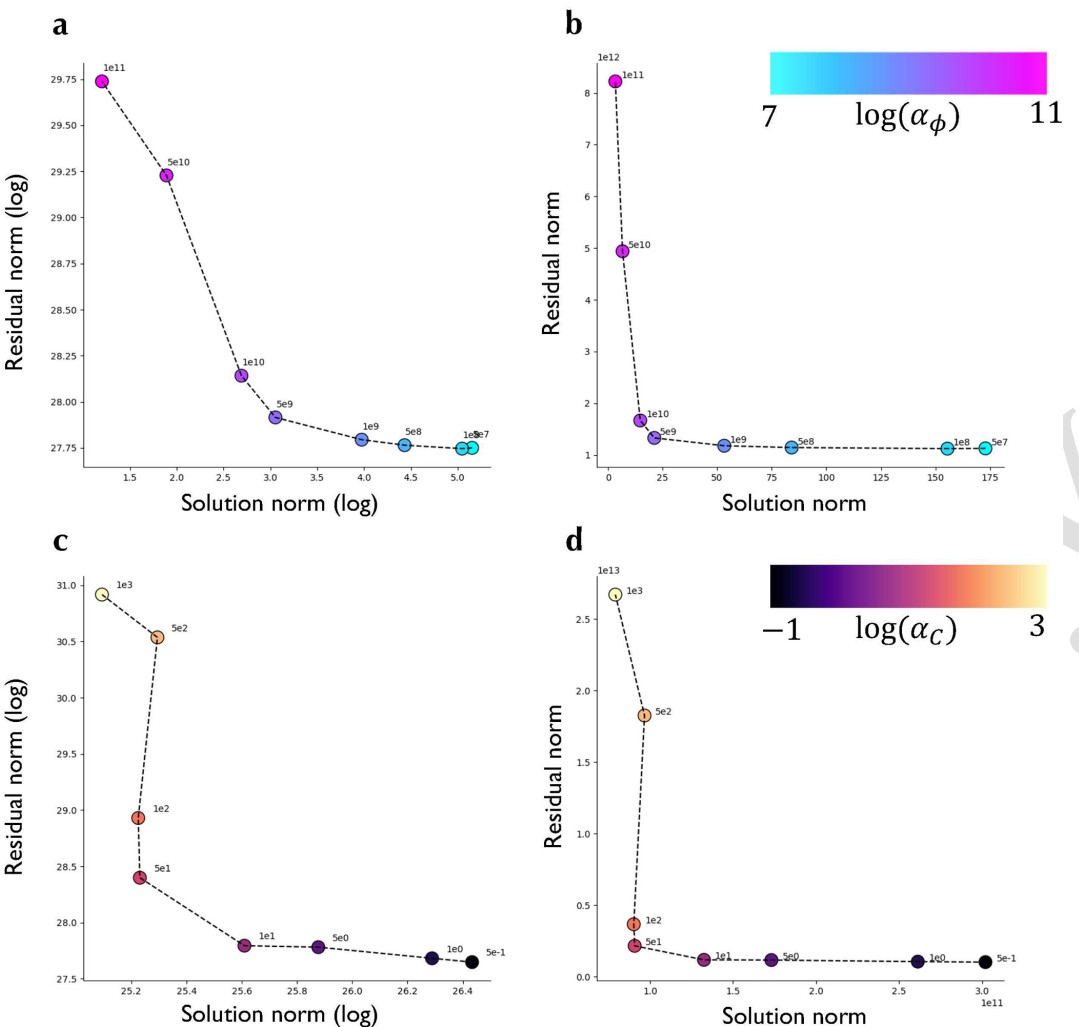

**Figure A1.** L-curves for the choice of regularisation parameters. **(a)** L-curve for $\alpha_\phi$. **(b)** Modified L-curve for $\alpha_\phi$. **(c)** L-curve for $\alpha_C$. **(d)** Modified L-curve for $\alpha_C$.

with a Weertman-like exponent of $m = 1/3$ and a threshold ice speed of $u_o = 300\,\mathrm{m\,a}^{-1}$ that represents a transition between viscous and plastic sliding (Joughin et al., 2019). This sliding law is physically plausible for the fast-flowing glaciers under consideration and allows for the greatest change in grounded ice speed with the small changes to resistive stress at the calving front brought about by the addition of sea ice. The value of $u_o$ is the main control on how far these speed changes propagate upstream of the grounding line.

To ensure the conclusions of this study are independent of the chosen $u_o$, we also consider a value of $u_o = 100\,\mathrm{m\,a}^{-1}$. For values of both 100 and $300\,\mathrm{m\,a}^{-1}$, the percentage difference in speed between the cases of no sea ice and 50 m of sea ice was calculated. We call these $\Delta_{100}$ and $\Delta_{300}$ respectively. Figure A2 shows the difference between these quantities.

We see that the difference is below 1 % across the HGE and Crane basins. The transects shown in Fig. 2f indicate changes in grounded ice speed with the addition of 50 m of landfast sea ice and a threshold ice speed of $300\,\mathrm{m\,a}^{-1}$ is on the order of 5 % where speed measurements were made.

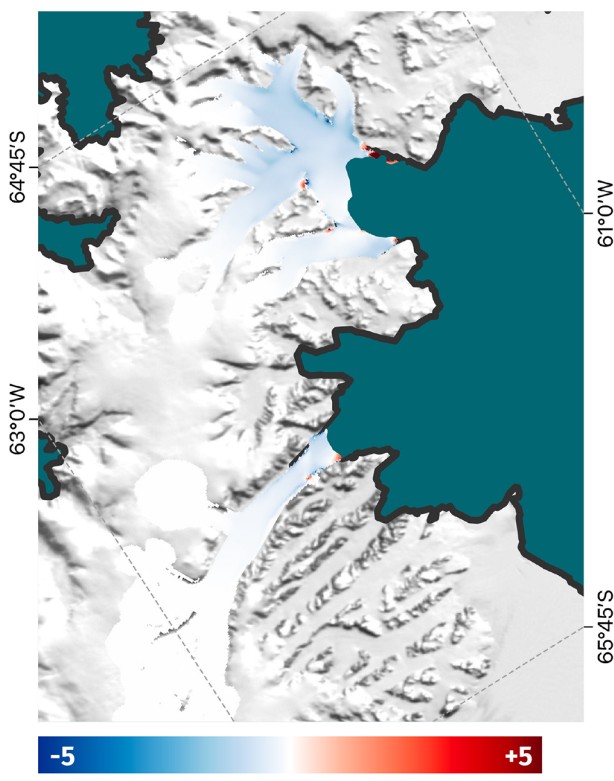

**Figure A2.** Difference between $\triangle_{100}$ and $\triangle_{300}$, i.e. the percentage change in ice speed with the addition of 50 m of landfast sea ice for $u_\mathrm{o} = 100\ \mathrm{m\,a^{-1}}$ and $u_\mathrm{o} = 300\ \mathrm{m\,a^{-1}}$ respectively. The background image is the MODIS MOA (Haran et al., 2021), and the black line is the glacier boundary according to Mouginot et al. (2017).

*Code availability.* The BISICLES ice sheet model is open-source and the code is available at https://commons.lbl.gov/display/bisicles/The+BISICLES+Adaptive+Mesh+Refinement+Ice+Sheet+Model (last access: 26 July 2022, Cornford et al., 2013).

*Data availability.* This paper is accompanied by the following datasets used in this study (DOI: https://doi.org/10.5281/zenodo.10580710, Surawy-Stepney et al., 2024): calving front and sea ice front positions; the mean speed and speed trend maps shown in Fig. 1; time series of ice speed shown in Fig. S2; quarterly flowline speeds shown in Fig. 2; freeboard and associated uncertainty point data shown in Fig. S5; maps of thickness and bed topography used in the model simulations; basal melt rates over the Scar Inlet; and shapefiles used for the extraction of speed, modelled speed, and sensitivity data.

*Supplement.* The supplement related to this article is available online at: https://doi.org/10.5194/tc-18-1-2024-supplement. TS2

*Author contributions.* TSS and AEH designed the work and wrote the paper. TSS led the modelling with contributions from SLC. AEH led the observations with contributions from TSS and the remaining authors. BJD processed ice speed data, analysed the ice speed data along with RAWS and SFW, analysed the atmospheric reanalysis data, and wrote the paper. BJW analysed the speed data and processed and provided grounding line locations. HLS and EKL provided calving front locations and fast sea ice extents. BIDF processed and analysed altimetry data across the glaciers. AR processed CryoSat-2 altimetry data, along with AS. LJ and NG produced and basal melt rate data over the Scar Inlet. All authors contributed to the scientific discussion, interpretation of results, and writing of the paper.

*Competing interests.* The contact author has declared that none of the authors has any competing interests.

ther geographical representation in this paper. While Copernicus Publications makes every effort to include appropriate place names, the final responsibility lies with the authors.

*Acknowledgements.* The authors gratefully acknowledge the European Space Agency (ESA) for the acquisition of CryoSat-2 data, as well as ESA and the European Commission for the acquisition and availability of Copernicus Sentinel-1 data.

*Financial support.* This research has been supported by the European Space Agency (grant nos. ESA-IPL-POE-EF-cb-LE-2019-834, ESA AO/1-10461/20/I-NB, and 4000128095/19/I-DT) and the Natural Environment Research Council (grant nos. NE/T012757/1 and NE/X019071/1).

*Review statement.* This paper was edited by Chris Derksen and reviewed by Jason Amundson and two anonymous referees. TS3

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

**Remarks from the language copy-editor**

CE1    Please give an explanation of why this needs to be changed. We have to ask the handling editor for approval. Thanks.

CE2    Please give an explanation of why the figure needs to be changed. We have to ask the handling editor for approval. Thanks.

CE3    Please give an explanation of why this needs to be changed. We have to ask the handling editor for approval. Thanks.

**Remarks from the typesetter**

TS1    Please give an explanation of why this needs to be changed. We have to ask the handling editor for approval. Thanks.

TS2    Please send a new supplement as a *.pdf without the title, authors, correspondence author, etc. as we will generate a supplement title page during publication (with a citation including the DOI), which will contain this information.

TS3    Please note that referees can decide on whether they want to be published as anonymous or under their name. Camilla Schelpe decided to be published as anonymous but was visible during the review process.