# Peer review of "The effect of landfast sea ice buttressing on ice dynamic speedup in the Larsen-B Embayment, Antarctica"

_The Cryosphere, 2023_

## Referee Comment (RC1)

[referee-annotated manuscript omitted]

---

## Referee Comment (RC2)

**Review of The impact of landfast sea ice buttressing on ice dynamic speedup in the Larsen-B Embayment, Antarctica, Surawy-Stepney et al., 2023**

This study focusses on the disappearance of the landfast sea ice from the Larsen B embayment in Jan 2022, and the impact this had on the glaciers that terminated in the region. The paper starts with an extensive summary of observations extracted from satellite data for the region, homing in on the Hektoria, Green and Crane glaciers that experienced the biggest speed up following the disappearance of the landfast sea ice.

Give the timing, it is most likely that the disappearance of the landfast sea ice caused the speed up of the glaciers in the region. However, the exact mechanism by which this occurred is uncertain. In this study, the authors are answering the valuable question of whether the sea ice conferred stability on the glaciers directly through a buttressing mechanism akin to the buttressing effect of confined ice shelves on upstream grounded ice.

They investigate both the direct buttressing of the grounded portion of the glaciers, and the buttressing of the floating ice tongues. The study uses a diagnostic numerical model to look at the flow speeds and stresses within the glacier, with and without an "ice shelf" of fixed thickness ranging from 1m to 50m, that represents the sea ice.

The conclusion of the paper is that the direct buttressing effect is too small to explain the observed disintegration of the ice tongues and so other mechanisms must be at play through which the sea ice conferred stability on the glaciers in the region. The authors provide many caveats on the limitation of the study such as the unknowns in the geometry.

I enjoyed reading the paper. The manuscript was well written, with a clear narrative that leads you through the paper, but I felt there were a few places where some more explanation is needed. Details below. I would fully recommend publication with these changes.

Specific Comments (in chronological order, not necessarily importance):

1. L4: "satellite measurements show that Hektoria, Green and Crane Glaciers have sped up by… more than 100 m a^{-1}." As it stands, it is not clear that these observations constitute part of the novel contribution of this paper, and yet Section 2 is devoted to the processing of the satellite data to arrive at these values. Perhaps add something like "*we show from* satellite measurements that Hektoria, …"

2. L79: "Speed changes extend up to 10 km upstream of the 2021 grounding line on Hektoria, Green and Crane Glaciers, where the speed up is most pronounced." Does this refer to the speed up being most pronounced on Crane glacier, or the 10km

upstream, or indeed on all three glaciers relative to the rest of the region? Consider rewording.

3. There are two places where the paper refers to "mean" ice speed, but it isn't clear whether that is data averaged over time, or a spatial average. It would be helpful for that to be made explicit.
    a. The first is the grey-scale plot in Fig 1.a "Inverse-error-weighted mean ice speed of glaciers… between October 2014 and April 2023". Is this the average over that entire time period? I found it surprising to have the average over such a long time, spanning the entire period that changes are being investigated in the study, if that is indeed what is plotted.
    b. The second is on L123: "mean observed ice speed across…in 2021". Is this what was plotted in greyscale in Fig 1a? Or is this an average over data gathered in 2021? Or in this case is it a spatial average?

4. L123. Add an explanation of what the enhancement factor \phi is in Biscicles. I don't think the authors define \bar{\mu} at any point in the paper (introduced on L502); presumably the viscosity? But the relationship to the enhancement factor is unclear. It would be helpful to explain how the rheology is set in the simulation.

5. L151-153. It wasn't immediately clear to me why this choice of Coulomb sliding law would ensure that "basal stresses on much of the grounded ice remain relatively unchanged". Could the authors elaborate?

6. L175-183, Fig 3 and Appendix A2. This section about the sensitivity to the ice thickness is quite difficult to make sense of and I think generally needs more explanation. It can feel a bit vague and confusing at points as it stands. I've added some specific points below:
    a. The label on Fig 3.a is "ln(Du/D\phi)". The caption says that (a) is log and (b) is linear. Does that mean the y-axis in Fig 3.b is "Du/D\phi"? It would be clearer to label the y-axis that way rather than simply "sensitivity", or define "sensitivity" in the caption.
    b. Could the authors expand on the derivation of Eqs (A3) and (A4) in the appendix? The reference to Goldberg & Sergienko 2011 is sufficiently different to the problem set up that it would be helpful to provide the full derivation here, and lay out the assumptions more clearly.
    c. Is the exact location of the coloured circles in Fig 3.a significant? My understanding from the text in the Appendix is that they mark the model domain \Omega_{HC} and represent the catchment area of the two glaciers, but reading the caption for Fig 3a it was quite confusing. "Magnitudes of different sensitivities of ice speed in the locations marked by the coloured circles…" It sounds as if those are two singular points. I would suggest either representing the domain \Omega_{HC} by an outline in the figure, or at least refer to "*regions* marked" not "*locations* marked" in the caption.

7. L192 – 199 and Fig 4. Could the authors elaborate on how they extracted the principal strain and stress components across the region? Is the direction for $\epsilon_1$ and $\sigma_1$ determined for each parcel of ice, or is it taken as the average for the domain?

8. The paper is well written overall, but Section 4 (Discussion) was generally a bit weaker than the rest of the paper, and lets it down. A few specific notes:
   a. Section 4.2 presents new data and comes as a bit of a surprise when reading the discussion section. Perhaps a more natural home for this section is under "Section 2. Observations"?
   b. L300: "Several studies have indicated the importance of sea ice…and the results of this study do little to suggest otherwise." The modelling results of this paper generally show that the buttressing action of the sea ice is not significant, so shouldn't this be "despite the results of this paper"?
   c. L302: "However, to more accurately judge the extent to which sea ice stabilises ice shelves… measurements at the critical zone near glacier calving fronts." This statement would be more meaningful if the authors gave specific detail about how access to these measurements would have helped their modelling study. What would different measurements enable you to do differently/more accurately in the study?
   d. L308: "We argue that the results represent an upper bound…. assuming viscous rheology, however, this may not be the case". I think given what follows, the authors mean that the results may not be restricted to *viscous* rheology, but it reads as if it may not be any kind of upper bound. Consider rewording that sentence.
   e. L310: "the specification of a particular constitutive relation has no impact on the stress distribution". I'm not sure what the authors are saying here; a different value for viscosity would certainly change the equilibrium profile of an ice shelf. Could this point be explained more?
   f. L311: "Fig. 3 a suggests that the sea ice in front of the centre of the ice shelf…has greatest impact on upstream flow." I assume this is referring to the magnitude of "ln(Du/D\phi)" in the plot, and yet by eye the values seem rather similar for all the sea ice in front of the ice shelf not just along the centre line. Could this be clarified?

Technical Corrections:

1. Fig 1(f). There is a mismatch between the figure legend and the caption, one has "HGE" and the other "Hektoria Glacier". I think in this case the time series does relate to Hektoria Glacier specifically so the legend on the plot should be updated?
2. Fig 3.b.1. The y-axis ticks are obscured by the label.
3. L136. "thin-ice-covered *leads*". I'd never heard this term before, is it a typo?

4. Fig 4a. The grey dashed line for the floating ice region of HGE does not line up with the edge of the reddish coloured domain. Is this just a plotting problem or is there something else going on here?
5. L216. "geomeotry" typo
6. L225. "to" missing
7. L316 "chocking" I'm not sure what that means in this context. Could it be a typo?
8. L538. "between" repeated.

---

## Referee Comment (RC3)

Review of Surawy-Stepney et al.: The impact of landfast sea ice buttressing on ice dynamic speedup in the Larsen-B Embayment, Antarctica

In this study the authors combine remote sensing observations and numerical modeling experiments to assess the potential impact of the loss of sea ice buttressing on the flow and stability of glaciers in the Larsen B Embayment. Essentially, they find that sea ice is unlikely to provide a direct control on glacier flow and stability, though they suggest that it can have indirect effects. Overall I think the paper is pretty easy to read and the results seem robust and interesting. That said, I do think the paper would benefit from moderate revisions. Here are a few general comments:

*Structure:* The paper is written in the way that one might tell a story, which isn't necessarily bad, except that I find it a little jarring to go back and forth between methods and results (particularly in Section 2). I understand that the methods in this section are relatively basic, but I still think a different structure here is warranted. Perhaps start with a paragraph or two that that describes all of the data sets and how you analyzed them before getting into a description of the observations. It also seems that there are a couple of other papers out there discussing similar observations. Make it clear how this study is different or complementary.

*Terminology:* The authors use the expression "floating ice/melange tongues". I don't know what this is referring to. "/" typically means "or", so is this "floating ice" or "melange tongues"? And what is meant by "melange tongues"? And later, is "ice tongue" really referring to an "ice shelf"? At least in the Antarctic context, when I hear about ice tongues I usually first think of something like the Drygalski Ice Tongue, which is not bounded by fjord walls.

*Sea ice model:* I have some concerns about the authors use of a viscous flow model to describe stresses in sea ice, and I'm not sure that I follow why this model should provide an upper bound on the buttressing stress. At the same time, I think the authors could use their observations and some simple arguments to support their conclusion that sea ice buttressing is not directly important unless the sea ice is tens of meters thick—which would also help to back up their model results.

Whether you invoke a viscoplastic or purely viscous rheology, the depth-averaged tectonic (or resistive stress) should scale with the depth-averaged strain rate:

$$R_{xx} \propto \dot{\epsilon}_{xx}, \tag{1}$$

where I am taking $x$ to be perpendicular to the glacier or ice shelf face and the tectonic stress is related to the Cauchy stress by

$$\sigma_{xx} = R_{xx} - P, \tag{2}$$

with $P$ the depth-averaged glaciostatic pressure. The force per unit width acting on the glacier face is then

$$F/W = -H\sigma_{xx} = -HR_{xx} + HP. \tag{3}$$

To get the buttressing stress, you need to subtract the force from the depth-averaged water pressure $P_w$, which would also act on the glacier if the sea ice was removed. This gives

$$F/W = -HR_{xx} + HP - HP_w = -HR_{xx} + \frac{1}{2}\rho g \left(1 - \frac{\rho}{\rho_w}\right) H^2. \tag{4}$$

The reason that it's not clear to me that a viscous model will provide a maximum bound on the buttressing force (as stated in line 145) is that I don't know how a viscous rheology will affect $R_{xx}$ compared to a viscoplastic rheology.

Nonetheless, the sea ice flow seems to be extensional in the observations, implying that $R_{xx}$ is positive. In other words, the last term in Equation 4 would seem to provide a good estimate of the upper bound on the sea ice buttressing force. Unless $H^2$ is large, this force will be pretty small.

Perhaps it would be interesting to compare the modeled buttressing force to the quasistatic force (i.e., when $R_{xx} = 0$).

One advantage of framing the discussion around something like Equation 4 is that it doesn't very strong assumptions about the rheology (e.g., which may be inconsistent with sea ice literature). You can also look at the field observations to get an idea of the forces involved without worrying about the details of the rheology. Only if the flow is highly compressive would I expect to see large forces. Perhaps that is happening at scales that you can't resolve in the satellite data, but then I'm not sure that they would be resolved in a viscous flow model either.

---

## Community Comment (CC1)

Comments on T Surawy-Stepney et al. ---
 "The impact of landfast sea ice buttressing on ice dynamic speedup in the Larsen-B Embayment, Antarctica"

N. Ochwat and T. Scambos

This paper discusses the extent of buttressing effects of recent Larsen-B embayment landfast sea ice and the consequences of its loss in 2022 on the tributary glaciers. It has great potential to contribute to the ongoing research regarding buttressing by multi-year landfast sea ice. Below, we present several short comments motivated by recent observations, which we hope the authors will consider when revising their manuscript.

- Regarding the methods section and the data used in the modelling, we agree with the reviewer 1 that more information is needed about the methods and data used. In particular, selected bedrock data as well as mapped location of the grounding zone may play an important role in the model's results. There are two primary options for bedrock data, Huss and Farinotti (2014) and BedMachine (Morlighem et al., 2023). Bedmachine data has recently been updated to incorporate sonar derived bedrock information for the terminus area of Crane, collected from the RV N.B. Palmer in 2006. Additionally, the Huss and Farinotti (2014) data do not cover the whole study area displayed in Fig. 1. What did the authors do in these regions?  Regarding the grounding line, it is unclear how this was determined and why it was chosen for the model; it would also impact how the glacier acceleration is discussed (i.e., floating or grounded ice). We note that the paper cited in support of the grounding line is not available at this time for review. It would be interesting to discuss the effect of using other, published, grounding lines (e.g. Rott et al., 2018, Sun et al. 2023, Ochwat et al. 2023).

- Furthermore, the model output of the "glacier terminus" (cyan lines figure 1) is likely incorrect as there is 300+ m thick ice several km further downstream from the glacier terminus identified in the model/figure 1. How are the authors defining the glacier terminus?

- What are the potential problems incurred by using a 2015 DEM with a 2021 Grounding line?

- In Figure 5, please include more years as this will help with understanding the timing of the events. We also request discussion on the fact that melt was lower in 2022 than previous years, according to AMSR data and that the largest peak in T2M was in 2021.

- The connection between basal melting and the near instantaneous break-up of the ice tongues after fast ice removal is unclear. Can the authors discuss how those two processes are connected? Also, do the authors have observation or model (reanalysis) indications of a large swell event leading to the breakup of the ice tongues? The

relationship between the basal melting, swell, and ice tongue response is generally unclear.

- We also would like to see the results discussed in the context of Robel, 2017.

  Robel, A. Thinning sea ice weakens buttressing force of iceberg mélange and promotes calving. *Nat Commun* **8**, 14596 (2017). https://doi.org/10.1038/ncomms14596

- The paragraph in Lines 326-336 is not clear.

- Lastly, the connection between the ice tongues' disaggregation and the loss of the fast ice needs to be better justified if the fast ice was not buttressing the tongues. The authors are implying an alternative mechanism for the instantaneous response if the fast ice was not offering any sort of buttressing effect, but the path to disaggregation is not clear in that case.

We look forward to seeing the revised manuscript.

---

## Author Comment (AC1)

**Responses to reviewer comments for the article "The impact of landfast sea ice buttressing in the Larsen-B Embayment, Antarctica"**

We would like to thank the editor and reviewers very much for the taking the time to read the article and for providing us with valuable and insightful feedback. All reviewer comments and my responses to each are collated in this document, so that each reviewer may see the others. Each review is reproduced here in full. Responses to any general comments of the reviewers are coloured in teal, while responses to specific comments are tabulated afterwards. A difftex file showing changes made to the manuscript is attached at the end of this document.

Following general comments by the reviewers I have made some changes to the structure of the article. Firstly, to help the flow of the article and reduce confusion around methods, there are some changes relating to the presentation of methods and results. There is a new "Observational methods" section listing the methods used in observations of sea ice extent and flow speed. The "Model set up" section has been renamed "Modelling methods" and contains a more complete description of the processes that went into making figures 2, 3, 4 and 6. I have also made changes to the structure of the discussion including adding the section "The use of a viscous flow model". This compares the stresses within the modelled viscous landfast sea ice, to the yield stress in a widely-used visco-plastic model, providing evidence that our experiments constitute a 'maximum buttressing example'. Various references to the choice of rheology scattered about elsewhere in the discussion have been removed. Additionally, I have taken the 'Environmental drivers' section out of the discussion. Finally, some of the discussion sections have been re-ordered. In line with some suggestions made in community comments, I have written more explicitly about mélange in the discussion and how it might relate to the buttressing effect of sea ice.

There was general agreement between the reviewers that it needed to be clearer what novelty there was in presenting our observations, given recent publications by others looking at this region. The revised manuscript makes it more explicit that the observations we present are complementary to concurrent studies, and exist in large part to provide specific background to the modelling study.

To reflect specific comments made by reviewers and community comments, I have made alterations to the majority of figures, though many alterations have been minor. Figure 2 no longer shows percentage speed change, and includes transects of speed observations for comparison with changes in modelled speed with the introduction of sea ice. Figure 4 has changed so that it no longer show percentage difference in principal strain rate but the original principal strain rate and the difference. I have added a new figure in the section "The use of a viscous flow model". I have added an additional supplementary figure showing the model thickness, speed, misfit and control fields following the model initialisation.

Finally, following a community comment which pointed out that the ice shelves were larger in reality than in our initial simulations, I have set up a new model geometry and performed the initialisation and simulations again. Each figure has been updated to reflect the results with the new geometry, though the changes have not affected the results of the article in any meaningful way.

**Responses to comments from Reviewer #2 (Camilla Schelpe)**

**Reviewer 2:** This study focusses on the disappearance of the landfast sea ice from the Larsen B embayment in Jan 2022, and the impact this had on the glaciers that terminated in the region. The paper starts with an extensive summary of observations extracted from satellite data for the region, homing in on the Hektoria, Green and Crane glaciers that experienced the biggest speed up following the disappearance of the landfast sea ice.

Give the timing, it is most likely that the disappearance of the landfast sea ice caused the speed up of the glaciers in the region. However, the exact mechanism by which this occurred is uncertain. In this study, the authors are answering the valuable question of whether the sea ice conferred stability on the glaciers directly through a buttressing mechanism akin to the buttressing effect of confined ice shelves on upstream grounded ice.

They investigate both the direct buttressing of the grounded portion of the glaciers, and the buttressing of the floating ice tongues. The study uses a diagnostic numerical model to look at the flow speeds and stresses within the glacier, with and without an "ice shelf" of fixed thickness ranging from 1m to 50m, that represents the sea ice.

The conclusion of the paper is that the direct buttressing effect is too small to explain the observed disintegration of the ice tongues and so other mechanisms must be at play through which the sea ice conferred stability on the glaciers in the region. The authors provide many caveats on the limitation of the study such as the unknowns in the geometry.

I enjoyed reading the paper. The manuscript was well written, with a clear narrative that leads you through the paper, but I felt there were a few places where some more explanation is needed. Details below. I would fully recommend publication with these changes.

We thank the reviewer for their praise of the article and their thorough and thoughtful review. Each suggestion has been taken into consideration and the vast majority have been implemented. My responses are tabulated below

**Responses to specific comments from Reviewer #2**

| ID | Reviewer Comment | Response |
|---|---|---|
| | Reviewer 2 | |
| 1 | L4: "satellite measurements show that Hektoria, Green and Crane Glaciers have sped up by... more than 100 ma$^{-1}$." As it stands, it is not clear that these observations constitute part of the novel contribution of this paper, and yet Section 2 is devoted to the processing of the satellite data to arrive at these values. Perhaps add something like "we show from satellite measurements that Hektoria, ..." | This is a good point. We show this to be true using ice speed measurements generated from satellite data, though the speed-up of these glaciers has been documented before in the preprint by Ochwat et al. (2023). We thought it best to include our independent measurements, made in parallel with Ochwat et al. (2023), though given the appearance of this other paper before our own, these observations do not constitute the major novelty of this work. Still, it is useful to remove the ambiguity, so I have added lines to the introduction making it clear that our observations are complementary to these other studies. I have also reworded this particular sentence as suggested to read: "We show using satellite measurements that, following a decade of gradual slow-down, Hektoria, Green and Crane Glaciers have sped up..." |
| 2 | L79: "Speed changes extend up to 10 km upstream of the 2021 grounding line on Hektoria, Green and Crane Glaciers, where the speed up is most pronounced." Does this refer to the speed up being most pronounced on Crane glacier, or the 10km upstream, or indeed on all three glaciers relative to the rest of the region? Consider rewording. | Thank you for pointing out this ambiguity. I meant the former, and have restructured the sentence to read: "On Hektoria, Green and Crane Glaciers, where the speed-up is most pronounced, speed changes extend up to 10 km upstream of the 2021 grounding line." |
| 3 | There are two places where the paper refers to "mean" ice speed, but it isn't clear whether that is data averaged over time, or a spatial average. It would be helpful for that to be made explicit.

a. The first is the grey-scale plot in Fig 1.a "Inverse-error-weighted mean ice speed of glaciers... between October 2014 and April 2023". Is this the average over that entire time period? I found it surprising to have the average over such a long time, spanning the entire period that changes are being investigated in the study, if that is indeed what is plotted.

b. The second is on L123: "mean observed ice speed across... in 2021". Is this what was plotted in greyscale in Fig 1a? Or is this an average over data gathered in 2021? Or in this case is it a spatial average? | Thank you for pointing out these points of confusion. In both instances, we are referring to averages taken over time.

a. This is indeed the average over the entire time period. It's only really meant to highlight the locations of the main glaciers to give the reader some understanding of how the area looks. The caption has been slightly reworded to make this seem less like the focus of panel a.

b. I have reworded this sentence to read: "We performed an inversion for basal traction ($C$) and stiffness ($\phi$) fields using observations of ice speed across the HGE and Crane basins averaged over 2021."

For reference, I have also replaced instances of 'enhancement factor' with 'stiffness' throughout the manuscript. |

| 4 | L123. Add an explanation of what the enhancement factor $\phi$ is in Biscicles. I don't think the authors define $\bar{\mu}$ at any point in the paper (introduced on L502); presumably the viscosity? But the relationship to the enhancement factor is unclear. It would be helpful to explain how the rheology is set in the simulation. | Thank you for pointing this out. I have added some general background to BISICLES ahead of the start of the new 'Modelling methods' section. This introduces the model, $\bar{\mu}$ and $\phi$ more completely. |
|---|---|---|
| 5 | L151-153. It wasn't immediately clear to me why this choice of Coulomb sliding law would ensure that "basal stresses on much of the grounded ice remain relatively unchanged". Could the authors elaborate? | The choice of sliding law is an important determinant of whether there will be any perceptible change in speed on grounded ice when the sea ice is added/taken away. The regularised Coulomb law is plastic above the threshold sliding speed. This speed, which we set to 300 m a$^{-1}$, is significantly less than the flow speed on most of the grounded ice we are interested in. This means that when the stress boundary condition at the edge of the ice shelf changes (because the sea ice is added or taken away), the new stress distribution has to be accounted for by changes in viscous stresses, i.e. changes in speed gradients, as basal stresses will remain the same. Using, for example, a linear sliding law, could suppress changes in speed. I have changed the sentences here slightly to read:
"For these simulations, we use a regularised-Coulomb sliding law (Schoof, 2005; Joughin et al., 2019) with a threshold ice speed of $u_o = 300$ m a$^{-1}$ so that sliding is plastic on much of the grounded ice. This ensures that basal stresses remain relatively unchanged as landfast sea ice is introduced. This results in enhanced changes to the viscous stress and, consequently, greater ice speed change." |

| 6 | L175-183, Fig 3 and Appendix A2. This section about the sensitivity to the ice thickness is quite difficult to make sense of and I think generally needs more explanation. It can feel a bit vague and confusing at points as it stands. I've added some specific points below: | Thank you for pointing out the confusing nature of the sensitivity plot and the discussion of how it was made. I have attempted to re-write the part of the "Modelling methods" section that deals with this to better explain what's going on. I hope that the reviewer finds the new manuscript less opaque. To deal with the specific points raised by the reviewer: |
|---|---|---|
| | a. The label on Fig 3.a is "ln($Du/D\phi$)". The caption says that (a) is log and (b) is linear. Does that mean the y-axis in Fig 3.b is $Du/D\phi$? It would be clearer to label the y-axis that way rather than simply "sensitivity", or define "sensitivity" in the caption. | a. I have changed the y-axes labels to read $\mathcal{D}u/\mathcal{D}\phi$ as suggested. |
| | b. Could the authors expand on the derivation of Eqs (A3) and (A4) in the appendix? The reference to Goldberg & Sergienko 2011 is sufficiently different to the problem set up that it would be helpful to provide the full derivation here, and lay out the assumptions more clearly. | b. Though it might be a bit unsatisfactory, I have not expanded on this derivation within the article, simply because it could become quite long. However, I have added perhaps a more appropriate reference that the reader can use to see how the calculation works (Morlighem et al., 2013). |
| | c. Is the exact location of the coloured circles in Fig 3.a significant? My understanding from the text in the Appendix is that they mark the model domain $\Omega_{HC}$ and represent the catchment area of the two glaciers, but reading the caption for Fig 3a it was quite confusing. "Magnitudes of different sensitivities of ice speed in the locations marked by the coloured circles..." It sounds as if those are two singular points. I would suggest either representing the domain $\Omega_{HC}$ by an outline in the figure, or at least refer to "regions marked" not "locations marked" in the caption. | c. This was not very well explained in the original article. The sensitivity is the gradient of the *mean ice speed over the areas shown by the coloured circles* with respect to $\phi$. I have tried to make this clearer in the modelling methods section, and by highlighting the areas in the revised figure. |
| 7 | L192 – 199 and Fig 4. Could the authors elaborate on how they extracted the principal strain and stress components across the region? Is the direction for $\epsilon_1$ and $\sigma_1$ determined for each parcel of ice, or is it taken as the average for the domain? | Good question, it is the former. The strain rate tensor was calculated local to each parcel of ice, in coordinate system aligned with the figure 4 a (x left to right along the bottom, y bottom to top). The principal strain rates were then calculated as local eigenvalues of the strain rate tensor with $\varepsilon_1 > \varepsilon_2$. Similarly for the principal stresses. The last sentence of the "Modelling methods" section now reads: "Principal strain rates $\varepsilon_1$ and stresses $\sigma_1$ (the largest eigenvalues of the strain rate and stress tensors local to each parcel of ice) were calculated for the HGE and Crane ice shelves for 10 m of lanfast sea ice vs 0 m.". |

| | | |
|---|---|---|
| 8a. | The paper is well written overall, but Section 4 (Discussion) was generally a bit weaker than the rest of the paper, and lets it down. A few specific notes:
Section 4.2 presents new data and comes as a bit of a surprise when reading the discussion section. Perhaps a more natural home for this section is under "Section 2. Observations"? | It is right to say that the discussion is a bit less focused than the rest of the article. I have made substantial modifications to the discussion in the revised manuscript and I hope that the reviewer finds these have improved it.
On this specific point, I have left the "Environmental drivers" section out of the observations section for the time being, but have given it its own section before the discussion. Moving it to "Observations" disrupts the flow of the main article slightly and makes these observations seem like too much of a focus of the article (whereas in reality they are not key to the narrative or conclusions of the article). We felt it was worth including these observations to highlight the many different processes going on at the time of the sea-ice evacuation and dynamic change in the glaciers. I have modified the section slightly - making it shorter and explaining the reasoning for its inclusion. |
| 8b. | L300: "Several studies have indicated the importance of sea ice. . . and the results of this study do little to suggest otherwise." The modelling results of this paper generally show that the buttressing action of the sea ice is not significant, so shouldn't this be "despite the results of this paper"? | This is a good point and I have added the caveat "- merely highlighting that this importance is unlikely to stem from its ability to buttress glaciers in the way an ice shelf can buttress a grounded ice stream." at the end of the sentence. |
| 8c. | L302: "However, to more accurately judge the extent to which sea ice stabilises ice shelves. . . measurements at the critical zone near glacier calving fronts." This statement would be more meaningful if the authors gave specific detail about how access to these measurements would have helped their modelling study. What would different measurements enable you to do differently/more accurately in the study? | I have added the example:
"For example, knowledge of sea ice thickness at the point of contact with the calving front, along with calving front morphology would help better our understanding of processes in which sea ice might inhibit iceberg calving via its influence on torques at the glacier front." |
| 8d. | L308: "We argue that the results represent an upper bound. . . . assuming viscous rheology, however, this may not be the case". I think given what follows, the authors mean that the results may not be restricted to viscous rheology, but it reads as if it may not be any kind of upper bound. Consider rewording that sentence. | This is a good point! Though this part of the discussion has been rewritten to no longer focus on the question of rheology in light of the new section "The use of a viscous flow model" that deals more completely with this issue. |

| | | |
|---|---|---|
| 8e. | L310: "the specification of a particular constitutive relation has no impact on the stress distribution". I'm not sure what the authors are saying here; a different value for viscosity would certainly change the equilibrium profile of an ice shelf. Could this point be explained more? | It was a mistake to include this line in the manuscript as it did not add much, while making things a bit confusing. What I was trying to get at is that there are 1D models of ice shelves where the stress distribution is uniquely determined by the geometry, regardless of the constitutive relation used. These don't extend to the situation under consideration but at the time of writing I wondered whether they did to some extent. This part of the discussion has been removed anyway in favour of the new section: "The use of a viscous flow model". |
| 8f. | L311: "Fig. 3 a suggests that the sea ice in front of the centre of the ice shelf... has greatest impact on upstream flow." I assume this is referring to the magnitude of "$\ln(Du/D\phi)$" in the plot, and yet by eye the values seem rather similar for all the sea ice in front of the ice shelf not just along the centre line. Could this be clarified? | As I have removed all the discussion of one-dimensional ice shelves, I have removed this line as well. |
| 9. | **Technical Corrections:**

1. Fig 1(f). There is a mismatch between the figure legend and the caption, one has "HGE" and the other "Hektoria Glacier". I think in this case the time series does relate to Hektoria Glacier specifically so the legend on the plot should be updated?

2. Fig 3.b.1. The y-axis ticks are obscured by the label.

3. L136. "thin-ice-covered leads". I'd never heard this term before, is it a typo?

4. Fig 4a. The grey dashed line for the floating ice region of HGE does not line up with the edge of the reddish coloured domain. Is this just a plotting problem or is there something else going on here?

5. L216. "geomeotry" typo

6. L225. "to" missing

7. L316 "chocking" I'm not sure what that means in this context. Could it be a typo?

8. L538. "between" repeated. | Thank you for pointing these out!

1. I have changed the legend to read "Hektoria". (The confusion occurred because the distance is drawn from a point on Hektoria, while the calving front is common the the whole HGE system.)

2. This has been fixed.

3. This is not a typo, I just mean leads in the sea ice (gaps of open water), covered perhaps with a layer of ice much thinner than the surrounding ice.

4. This is not a plotting problem, the domain was drawn this way to avoid values very near the front that may be wrong due to edge effects when we differentiate the velocity components.

5. This has been corrected.

6. This has been corrected.

7. Perhaps a better term would have been 'wedging'? Regardless, this line has been removed anyway.

8. This has been corrected. |

**Responses to comments from Reviewer #3 (Jason Amundson)**

**Reviewer 3:** In this study the authors combine remote sensing observations and numerical modeling experiments to assess the potential impact of the loss of sea ice buttressing on the flow and stability of glaciers in the Larsen B Embayment. Essentially, they find that sea ice is unlikely to provide a direct control on glacier flow and stability, though they suggest that it can have indirect effects. Overall I think the paper is pretty easy to read and the results seem robust and interesting. That said, I do think the paper would benefit from moderate revisions. [Below] are a few general comments.

We thank the reviewer for their praise of the study and insightful comments. Their feedback has been very valuable and, I hope they agree that the resulting changes have resulted in a clearer and more robust article.

**Structure** The paper is written in the way that one might tell a story, which isn't necessarily bad, except that I find it a little jarring to go back and forth between methods and results (particularly in Section 2). I understand that the methods in this section are relatively basic, but I still think a different structure here is warranted. Perhaps start with a paragraph or two that that describes all of the data sets and how you analyzed them before getting into a description of the observations. It also seems that there are a couple of other papers out there discussing similar observations. Make it clear how this study is different or complementary.

Considering the reviews together, it is clear that the article would benefit from a restructure in the way the reviewer suggests. I have fit section 2 into a more traditional structure with separate 'Observational methods' subsection before going into the two 'Sea ice area change' and 'Ice dynamic and calving response' results sections. The modelling sections have been adjusted with an extended "Modelling methods" section followed by results. I hope the reviewer finds the new structure easier to read.

Regarding how the observations presented in this article differ from those in concurrent studies, I have not made substantial changes detailing where results are complementary or conflicting. As a bit of a catch-all, I have added the following sentence to the introduction:

"These results complement recent work by Sun et al. (2023) and Ochwat et al. (2023), which document many similar observations to those presented here."

I think it is useful to present our observations as they differ to some extent and form a more specific background to the modelling study. (This also allows us to do things like compare the modelled and observed speed changes at the same locations.)

**Terminology:** The authors use the expression "floating ice/melange tongues". I don't know what this is referring to. "/" typically means "or", so is this "floating ice" or "melange tongues"? And what is meant by "melange tongues"? And later, is "ice tongue" really referring to an "ice shelf"? At least in the Antarctic context, when I hear about ice tongues I usually first think of something like the Drygalski Ice Tongue, which is not bounded by fjord walls.

I didn't commit fully to using the term 'ice shelves' because, for a while at least, there was no real boundary between the ice shelves and the mélange in front of them. However, this is not very clear and it is certainly correct to refer to the bulk of the floating extensions of the glaciers as "ice shelves". Hence, I have reverted to more standard uses of these words. I have removed all instances of "ice/mélange", changed all instances of "ice tongues" to "ice shelves". At various points in the article, it is made clear that there are regions of mélange in the transition between the pure ice shelves and the pure landfast sea ice.

**Sea ice model:** I have some concerns about the authors use of a viscous flow model to describe stresses in sea ice, and I'm not sure that I follow why this model should provide an upper bound on the buttressing stress. At the same time, I think the authors could use their observations and some simple arguments to support their conclusion that sea ice buttressing is not directly important unless the sea ice is tens of meters thick—which would also help to back up their model results.

Whether you invoke a viscoplastic or purely viscous rheology, the depth-averaged tectonic (or resistive stress) should scale with the depth-averaged strain rate:

$$R_{xx} \propto \dot{\epsilon}_{xx} \tag{1}$$

where I am taking x to be perpendicular to the glacier or ice shelf face and the tectonic stress is related to the Cauchy stress by

$$\sigma_{xx} = R_{xx} - P \tag{2}$$

with P the depth-averaged glaciostatic pressure. The force per unit width acting on the glacier face is then

$$F/W = -H\sigma_{xx} = -HR_{xx} + HP. \tag{3}$$

To get the buttressing stress, you need to subtract the force from the depth-averaged water pressure $P_w$, which would also act on the glacier if the sea ice was removed. This gives

$$F/W = -HR_{xx} + HP - HPw = -HR_{xx} + \frac{1}{2}\rho g \left(1 - \frac{\rho}{\rho_w}\right) H^2. \tag{4}$$

The reason that it's not clear to me that a viscous model will provide a maximum bound on the buttressing force (as stated in line 145) is that I don't know how a viscous rheology will affect $R_{xx}$ compared to a viscoplastic rheology.

Nonetheless, the sea ice flow seems to be extensional in the observations, implying that $R_{xx}$ is positive. In other words, the last term in Equation 4 would seem to provide a good estimate of the upper bound on the sea ice buttressing force. Unless $H^2$ is large, this force will be pretty small.

Perhaps it would be interesting to compare the modeled buttressing force to the quasistatic force (i.e., when $R_{xx} = 0$).

One advantage of framing the discussion around something like Equation 4 is that it doesn't very strong assumptions about the rheology (e.g., which may be inconsistent with sea ice literature). You can also look at the field observations to get an idea of the forces

involved without worrying about the details of the rheology. Only if the flow is highly compressive would I expect to see large forces. Perhaps that is happening at scales that you can't resolve in the satellite data, but then I'm not sure that they would be resolved in a viscous flow model either

The reviewer is right to raise this very important point. The conclusions of the article rest heavily on the assertion that the viscous flow model provides an upper bound on the buttressing capacity of the ice (far more than, for example, the accuracy of the geometry or the order of the adding of the sea ice in the model vs in reality). We have taken on board the insightful points raised and useful suggestions in this part of the review. However, I would argue that rather than making substantial changes to the methods used in the study, all that is needed is the addition of some more explanatory text.

Firstly, I agree with the reviewer's analysis and the resulting equation 4. This, along with the suggestion of looking at extensional or compressive flow would be a good solution when considering a one-dimensional model. However, the difficulty in the system under consideration is that it is two-dimensional, where it is difficult to argue that a local measure of buttressing such as that proposed by the reviewer would accurately describe the effect of the sea ice on the glacier ice. Instead, things like shear stresses near the margins of the sea ice, and the lateral transmission of the stresses are likely to be critical. Though we could estimate shear strain rates from observations of sea ice flow speed, it would be difficult to assess the net effect of these on the glacier.

Regardless, the existence of other ways of looking at the problem does not invalidate the method we chose to apply; but our choice of rheology could - as the reviewer is right to point out. However, there are good reasons believe that the use of a viscous flow model is sufficient to answer the question posed in the article about sea-ice buttressing. I have added a variant of following argument, along with an additional figure (reproduced below - Fig. 1) to a new section of the discussion called "The use of a viscous flow model" that I hope makes this more convincing. In essence, it shows, as the reviewer asks, how a change in the rheology might affect resistive stress (though we consider viscous stress).

Firstly, it is clear that there is large-scale deformation over the landfast sea ice in the embayment, above the random wedging together of floes. A number of different rheologies have been considered for sea ice, but it is generally agreed that large-scale deformation is plastic - due to the opening of cracks, raising of pressure ridges and shearing along crack boundaries. The stress state for such deformation lies on a yield curve of critical stresses. Various commonly used models employ this kind of rheology. An assumption of isotropic ice allows this yield curve to be plotted as a function of principal stresses. However, the models disagree on subcritical rheology - where the bumping of floes together and deformation internal to individual floes is important. Intuitively, this might be first considered as elastic due to the short-timescales on which these kinds of interactions happen (Coon et al. (1974)). However, it was shown by Hibler (1977) that the jostling of floes approximates viscous behaviour at small strain rates. This led to a commonly used parametrisation of sea-ice rheology that takes the form:

$$\sigma_{ij} = 2\eta\dot{\varepsilon}_{ij} + [(\zeta - \eta)\dot{\varepsilon}_{kk} - \frac{P}{2}]\delta_{ij} \tag{5}$$

where $P$ parametrises the strength of the ice and $\zeta \equiv \zeta(\dot{\varepsilon}_{ij}, P)$ and $\eta \equiv \eta(\dot{\varepsilon}_{ij}, P)$ are bulk

and shear viscosities respectively. The functional forms of the viscosities, along with the specification of an eccentricity, determine the shape of a yield curve, passing through the origin, on which the stress state lies at typical strain rates. The ice strength in Hibler (1979) is approximated as:

$$P = P^* h A e^{-c^*(1-A)} \qquad (6)$$

where $P^* = 27.5$ N m$^{-2}$, $c^* = 20$, $h$ is the sea ice thickness and $A$ is its concentration. $P^*$ is sometimes treated as a tunable parameter but $P$ is most often within a factor of 10 of $10^5$ N m$^{-1}$ (Feltham (2008)). Regardless of the precise sub-critical rheology, the strength parameter $P$ is the key scale for stresses that can be maintained within the sea ice.

[Figure]

Figure 1: A comparison of viscous stresses in the modelled landfast sea ice and a possible yield curve for sea ice. The graph shows viscous stresses in the modelled landfast sea ice plotted as a function of "negative pressure" $\sigma_I$ and "maximum shear stress" $\sigma_{II}$ (Feltham (2008)). The colour of the points indicates where in the domain the modelled viscous stress is extracted - corresponding to regions on the inset map. The grey circle on the graph indicates a yield curve for a possible sea ice rheology of the form of eq. (5) with $P = 10^6$ N m$^{-1}$ and eccentricity of 0. The grey dashed ellipse shows a yield curve with eccentricity of 2 and $P = 2.75 \times 10^5$ N m$^{-1}$ as suggested in Hibler (1979).

We plot compressive and shear stress invariants within our 10 m-thick viscous sea ice (Fig. 1). We show on the same scale a possible but very generous yield curve for isotropic sea ice that is as strong in shear as it is in compression (this is not true and a real yield curve would be squashed vertically within the circles) with a strength of $10^6$ N m$^{-1}$. (We also plot a more realistic yield curve with eccentricity of 2 and $P = 2.75 \times 10^5$ N m$^{-1}$ suggested in Hibler (1979).) This shows that in the small embayments local to both the HGE and Crane glaciers, the viscous sea ice in our model holds a considerably larger amount of stress than is thought to be possible in real sea ice. This results in a higher buttressing effect.

**Responses to general comments from Reviewer #1**

We thank the reviewer for taking the time to read and comment on the article and for their valuable feedback. Their review is split into two components - a written part giving general feedback, and various specific comments in an annotated pdf of the manuscript. I have transcribed their annotations into the table at the bottom of this document.

**Reviewer 1:** This paper presents satellite-derived glacier speeds from 2014 and landfast sea ice extent from 2002 till present. A numerical model is used to compute whether landfast sea ice (modeled as a thin meteoric ice shelf) could have provided sufficient buttressing to produce the observed glacier speedup. The answer to this question is no, following a standard modeling approach based on diagnostic runs quantifying instantaneous changes in glacier speed resulting from different boundary conditions. Additional hypothesis are then offered regarding potential causes of the glacier speedup, but these are no longer supported with numerical modeling.

**Satellite estimates:**
There are a lot of statements throughout the paper about pre/post 2011. But something that hasn't been shown is what the glaciers and glacier fronts did in terms of speed between 2002 and 2011 - how quickly did the system stabilize past the fast speed up in 2002? Was there any change in glacier speed around 2011 when the landfast sea ice settled in? And did the ice tongues only form past 2011 as mentioned in the paper? I don't think any of that was shown here, but is important for understanding the role of sea ice in the glacier dynamics in this bay.

The paper talks in a lot of detail about trends from 2014 but not clear what that reflects. Is that still recovering from 2002 speedup? is it reacting to 2011 landfast sea ice presence?

These are all very interesting points to consider! I agree that a greater analysis of the changes to glacier speed and calving front position covering the period before and after initial sea ice growth would provide greater insight into the effect of the sea ice. I have included some additional references to the literature that covers a period spanning pre- and post-2011 (namely: Wuite et al. (2015) and Rott et al. (2018)). However, I have not expanded very much on these points as the aim of the article is not really to fully review the effect of the landfast sea ice, rather to address the concept of sea ice buttressing. Similarly, we are not particularly concerned with what caused the glacier speed-up after the disintegration of the landfast sea ice in 2022. Further, this article does not presume to completely describe the mechanisms involved in the post-2014 trends, but does suggest as motivation for the modelling study that landfast sea ice can promote the growth of ice shelves that act as a control on upstream flow.

Previous papers have already shown similar satellite-derived information as shown here (and the studies are cited in this manuscript), so I wasn't clear on what the novelty here was. Making that explicit could help.

This is true - the novelty of the work does not lie greatly in the observations, other than them being slightly more focused on the HGE and Crane systems, as well as providing slightly more detail on the spatial pattern and magnitude of speed change. This was a comment raised by reviwer #3 as well. There is significant overlap between the observations presented here and in pre-prints elsewhere in the literature. However, we thought it was useful to present our additional observations as background and motivation to our modelling study. (This also allows us to do things like compare the modelled and observed speed changes at the same locations.) I have added the following sentence to the introduction:

"These results complement recent work by Sun et al. (2023) and Ochwat et al. (2023), which document many similar observations to those presented here."

**Modeling:**

There is no model validation, although Larsen B offers a rare opportunity in glaciology to actually validate a model in some sense. If the consensus is, that the Larsen B ice shelf breakup caused acceleration of outlet glaciers (and some models have showed that using similar techniques as here), then I would expect to see that validation here. That is, to see the model in this study first show that the removal of the ice shelf from a model tuned to pre 2002 velocities reproduces observed post 2002 speedup. If it does so, then I think the conclusions about the relative insensitivity of the land ice to landfast sea ice buttressing will be more robust.

The suggestion that there is no model validation is not entirely correct - this study is an exercise in validation. We ask the question of whether this land ice + sea ice model is able to reproduce the observations of speed change and the answer is no. However, I can in principle see the value in the kind of validation that is being suggested (with an experiment looking at the removal of the Larsen-B ice shelf). It might answer whether the geometry is set up correctly, or whether the sliding law we use is plausible. However, this does not have a great deal of relevance for our study as the geometry has changed, and we seek a sliding law that maximises the impact of the sea ice. More importantly, if we carried out that kind of validation, I think we would be limited to diagnostic modelling procedures like the ones we present here. I refer the reviewer to Figure 2 showing linear changes in ice speed with sea ice thickness to show that the model produces glaciers that are sensitive to changes in boundary stress - like a validation using the Larsen-B Ice Shelf would no doubt suggest.

There is an inconsistency in the modeling. The inversion uses ice flow speeds from 2021 when landfast sea ice was present, but the effect of the landfast sea ice itself is not included in the model at this point. So the tuning to velocities is done with the wrong setup/geometry. Further, it is unclear why the authors do the opposite experiment to nature. They take an (inconsistently) tuned state and then look at the effect of addition of an ice shelf, rather than the effect of the removal of an ice shelf. At the very least it would be good to explain the reasoning for that choice and an argument for why this procedure is generally reversible (especially considering potentially long lasting transients).

This is an astute point, and one that could be dealt with better in the article. Firstly, the fact we see no noticeable change in glacier flow with the addition of sea ice makes

the solution to the inverse problem in the case *without* the sea ice an almost equally good solution to the case *with* the sea ice. Hence, differentiating between the model initialisations in the two cases is not particularly meaningful. Taken together with Fig. 2d2 (now Fig 2f) (showing changes in speed to be linear in sea ice thickness) which shows that the addition of small amounts of sea ice cause changes comparable to taking away small amounts of sea ice, it is clear that the results would be the same had the experiment been performed the other way round. Had that been done, the experiments would just have taken substantially longer and perhaps been more difficult. It is true that the choice to do the experiments this way round assumes from the outset that the sea ice will not affect the glacier flow, but this assumption is validated by the results. I have added the following couple of sentences in the 'Modelling methods' section:

"The choice to do this presupposes that the inclusion of the landfast sea ice will do little to change the solution to the inverse problem and is necessary as the thickness of the landfast sea ice is not well constrained. We shall see that this assumption is validated by the results."

An alternative argument for why this consideration is not relevant also answers questions regarding modifications to the modelled glacier geometries. The argument we make in the paper is essentially a mechanical one: adding a thin layer of landfast sea ice in front of the glaciers has little impact on the glacier flow. It would take a considerable conspiracy of factors for this to not be true given slightly different solutions for the model control parameters $C$ and $\phi$, or, for example, the position of the grounding line, or the length of the ice shelves.

The modeling emphasis is on the glaciers that showed large change in observations, and the statement is that the modeled response after sea ice removal is not enough. How about the response of glaciers that showed no change in observations? Is the response to the sea ice removal in model larger than the observed change? Maybe you show something for Evans in a figure but there is no mention of this at all in the text. This is another opportunity to have a "control", or a validation of your model and setup.

Thank you for this suggestion. I have highlighted this by expanding the following line in the first paragraph of the section "Direct buttressing of grounded glaciers":

"These modelled percentage changes in speed are similar in magnitude on all glaciers including Evans, where we do not observe a substantial dynamic response (Fig. 1 g)". This is an indication in its own right that landfast sea ice removal was not the primary cause of the ice dynamic change observed on these glaciers.

3.2.2 talks about sea ice buttressing effect near the termini, but nowhere it is shown whether the (observed) loss of the ice tongues produces the observed grounded ice acceleration. I think the paper could go a bit further with the modeling experiments and show whether the ice tongue loss does reproduce the observations of speedup.

I think this would be a nice thing to do, though not here as the article is not particularly concerned with what caused the glacier acceleration. I am happy to leave the relationship between terminus retreat and speed change suggested but not fully explored.

In the end a hypothesis is suggested that basal melt rate change could have contributed to the glacier speed up - can you show with a model that the inferred change

in melt rates over the duration of the change can indeed produce significant change in glacier speed over the observed time period?

I did not intend for this statement to be seen as much more than a possible suggestion that highlights the many different forcings at play in the region. I hope that it is a bit clearer given the modified structure that this is not central to the article, and hence would not require its own modelling study.

**Methods:** There is minimal methods section so it is sometimes quite hard to assess what the authors actually did.
For example, which points they chose to show time series, how representative those points are, what do the error bars (shaded ares) actually represent, etc.

It is clear from the reviewer responses in general that the article was not structured in the best way. To improve the readability, I have added a new "Observational methods" section that includes this extra information, as well as a fairly comprehensive "Modelling methods" section.

The sections 3.2.1 and 3.2.2 need to match corresponding sections/paragraphs in "Modeling setup" so that it is clear which modeling experiment will answer what question/hypothesis.

This is a good suggestion that should help the reader with which methods were used for which experiments. This has been implemented in the new "Modelling methods" section. However, rather than separating out methods sections corresponding to sections 3.2.1 and 3.2.2, I have added sentences making it clear which methods correspond to which results sections. Explanations of the specific experiments have been moved from sections 3.2.1 and 3.2.2, leaving these sections to focus on the results.

**General:** The study really tries to be concise but that is at the expense of clarity and detail. At many places justification of choices and any sort of reasoning is completely absent.

I hope that the new methods sections make things slightly clearer and that, in general, the reviewer finds the revised manuscript easier to read.

The paper tries to merge observations with modeling but doesn't succeed in joining the two together very well. Part of it is probably the presentation. It needs to be stated a bit more clearly what the role of the model is in this study and what exact question, provided by the observations, it aims to answer. And also how the model and observations complement each other. Part of this is probably the inconsistency between the time series from observations, but then diagnostic model, rather than time evolving one is used. I understand that this is what people have been doing now for a while and it is fairly standard, but at some point it would be good to start understanding transient responses.
A good example of the lack of joining model and observations together is that the modeling results are in no way compared visually to observations (e.g. in Fig 2, could you also add observed flow speeds before and after sea ice removal, in addition to the modeled ones?)

I have tried to make it clearer in the revised manuscript that the study aims to answer whether sea ice buttressing was a factor in the change in glacier dynamics seen after the sea ice evacuation, and that the observations provide motivation for this question. Regarding the simulation of transient responses, though in an ideal world we would carry these out, there are a couple of reasons why it is not appropriate here to do so. The most important of which is that we do not have an appropriate calving law. Hence, the only real ways of isolating the effect of the landfast sea ice buttressing given the large calving events that took place in the region are through the kind of basic mechanical arguments we make here.

It is a good suggestion to make the modelled and observed speed changes more visually similar. I have changed figure 2 in the way the reviewer has suggested. The figure is simplified, with no percentage speed changes shown, and ice speed data for the second quarter of 2020 and the last quarter of 2022 is plotted along with the modelled speeds.

The language is a bit clumsy, some long, confusing (ambiguous) sentences are present. Often times unclear what authors mean, some repetition is probably necessary.

I have tried to improve this in the revised version.

Terminology is also an issue, often times the authors use many different terms to refer to the same thing, introducing ambiguity. Also, when saying ice, in this particular study it is really important to specify every time whether you mean sea ice/meteoric ice.

This was an issue also raised by reviewer #2 - particularly regarding uses of 'ice' vs 'mélange' and 'ice shelf' vs 'ice tongue'. I have tried to commit more to particular vocabulary. I hope the reviewer finds the terminology in the modified manuscript less ambiguous.

**Responses to specific comments from Reviewer #1**

| Reviewer 1 | | |
|---|---|---|
| ID | Reviewer Comment | Response |
| 1 | Suggested change in title: effect, rather than impact is probably more accurate. "impact" suggests it is strong | I have implemented the change suggested by the reviewer. |
| 2 | Suggested change in abstract: again, effect | I have also implemented this change, and on a few other occasions throughout the article. |

| 3 | Line 12: "However, as the accompanying changes in viscous stress were small compared to local spatial variation":
 I'm not sure what you mean here at all | I apologise for the confusing sentence. I have replaced this with: "However, the accompanying changes to the distributions of viscous stress within the ice shelves were small compared to the widths of those distributions. Hence, this loss of buttressing is likely to have been a secondary process in the disintegration of the ice shelves compared to, for example, increased ocean swell or the same factors that brought about the initial landfast sea ice disintegration." |
|---|---|---|
| 4 | Line 34: citation of Rott et al., 2018:
 I don't understand how a 2018 paper can be used for a statement refering to 2022 desintegration | I have put the citation before the word "disintegration" (it was meant to refer to the growth of the ice shelves, rather than their disintegration post sea-ice evacuation but was in slightly the wrong place.) |
| 5 | Figure S1:
 The colors in S1 are really a bit too close to each other in tone to be able to tell which line is which. Multiple panels by 5 year segments, or additional timeseries of ice front positions along some lines are plausible ways how this could be visualized with better clarity | I have implemented this suggestion, and replaced Fig. S1 with one split into 4 panels, each with 5 years of sea ice front locations. |
| 6 | Line 49 "... sea ice was retained throughout the summer months"
 I don't think that is what Fig. 1a shows. It has sea ice edge from November (December) each year, but austral summer ends in February or so | The sentence at the start of the section "Sea ice area change" has been modified. |
| 7 | Line 49 "... and generally grew in extent each year through to 2017 when..."
 Again this is not clear from the figure, it seems it is more less staying the same in 2014-2016 | Thank you for pointing out this inconsistency, this has been rectified in the modified article. |
| 8 | Line 51 "Between the 18th and 23rd of January 2022, the multi-year sea ice disintegrated..."
 provide reference | I have added a citation to Ochwat et al. (2023) here. |
| 9 | Line 51 "Satellite data show that there was modest surface melt ponding on the sea ice during the austral summers prior to the sea ice collapse, however, these melt ponds were more widespread and densely spaced across the entire sea ice area in 2021. Surface melt ponds were observed to be at their maximum extent in December 2021 immediately prior to the sea ice disintegration in January 2022."
 Is that shown here or elsewhere? Please clarify or provide reference | These lines have been removed from the manuscript. Ochwat et al. (2023) discuss the causes of the sea ice disintegration fairly extensively, and discussion of this at this point in the manuscript does little to help motivate the modelling. |

| 10 | Line 75 "At the grounded ice locations chosen for extraction of speed timeseries, ..."
 I think you need to comment a bit more on how you chose these locations and how representative they are | It is a good suggestion to make it clear in the article that nearby locations show similar signals. It should be clear from Figs 1b-c, which show trends that are compatible with the timeseries data nearby to the points we picked, and the new Figure 2, that the changes are more widespread than the locations we chose. I have added a sentence to the new "Observational methods" section explaining the choice: "These locations were chosen to be in the centre of the grounded ice streams, close to the grounding line but with enough room for a 1 km buffer along the flowlines that pass through them." |
|---|---|---|
| 11 | Referring to the "sign change" mentioned on line 77:
 This seems very small | I agree this signal is weak. I have softened the sentence a bit to say a "potential sign change", which is then qualified in the remainder of the sentence as before. |
| 12 | Referring to the "sign change" mentioned on line 77:
 do you mean sign change in acceleration? | We say "sign change in the ice speed trend" rather than "sign change in acceleration" to make it clear that we are looking over longer timescales. |
| 13 | Line 80 "...where the speed-up is most pronounced":
 I don't see where that is shown | This sentence was ambiguous, I did not mean that speed-up on was most pronounced at the grounding line, rather that speed-up was most pronounced on Hektoria, Green and Crane Glaciers. I have changed the sentence to read:
 "On Hektoria, Green and Crane Glaciers, where speed-up is most pronounced, we see speed changes extend up to 10 km upstream of the 2021 grounding line." |
| 14 | Line 81 "Overall, the speedup observed after the sea ice disintegration in January 2022 is more extensive than the region of slowdown observed between 2015 and 2022 (Fig. 1 d), and extends inland onto the grounded ice sheet (Fig. 1 e), therefore increasing the rate of ice discharge into the ocean":
 I am not sure what you are trying to say here. Are you comparing total ice discharge over the slowdown years with ice discharge over the much shorter period of speedup and claiming the latter is higher despite the short duration? If so, please clarify | This is a bit of an odd sentence but I think it does make sense - we are comparing the '*rate* of ice discharge' not the integrated discharge. But it's not particularly relevant so I have removed the sentence. |
| 15 | Figure 1
 Figure 1 caption is really long. Probably best to split the manny (unrelated) panels to different figures, each with a brief caption, for clarity and readability. | I have shortened the caption but have kept the figure together for the time being. |
| 16 | Figure 1 f:
 the y axis on the actual panel says something different than what the description is here. Also, from the y label it is unclear what max is | I have changed the y-axis on the panel to read "Distance to the calving front (km)" |

| 17 | Figure 1 f - caption "Timeseries of distance to the calving front from a point on Hektoria Glacier...": Which point? | I have removed this reference to 'a point' on the glacier, and left the sentence saying the distance is measured along the white lines. |
|---|---|---|
| 18 | Line 89 "Hektoria and Green Glaciers were further exposed...": To what? | Good point, I'm not really sure what that means specifically. I have replaced this line with: "The HGE Ice Shelf retreated by a further 9 km between September and December 2022, decoupling the ice shelves of Hektoria and Green Glaciers." |
| 19 | Line 91 "The observations of the Larsen-B Embayment presented in this study and others (Ochwat et al., 2023) suggest that landfast sea ice permitted the growth of the floating ice/mélange tongues in front of HGE and Crane Glaciers over the period 2011 to 2022, which acted as a control on the upstream glacier dynamics.": I missed where you showed the glacier ice front position over the 2002-2022, so that we can see that the growth of ice tongues only occured after 2011 - or was that shown in a different paper? (Ochwat doesn't show that) | This is not explained very clearly in the article, so thank you for pointing out the confusion here. I have changed the first few sentences of the section 'Landfast sea ice buttressing' to read: "Previous studies focused on the area have shown ice speed changes on HGE and Crane Glaciers to be concurrent with changes in terminus position (Wuite et al., 2015; Rott et al., 2018) prior to 2011. In the case of HGE these changes fluctuated, while on Crane steady terminus advance accompanied steadily decreasing glacier speeds. Following the growth of persistent landfast sea ice in 2011, we see persistent terminus advance and decreasing speed - show here and elsewhere (Wuite et al., 2015; Rott et al., 2018; Ochwat et al., 2023). These observations, along with that of ice shelf disintegration after the sea ice evacuation in 2022, shown here and in Ochwat et al. (2023), suggests a coupling of landfast sea ice to glacier dynamics in which landfast sea ice permitted the growth of the ice shelves in front of HGE and Crane Glaciers prior to 2022 which acted as a control on the upstream flow. " |
| 20 | Line 105 "sea ice": maybe you should stick to the term landfast sea ice whenever that is the type of sea ice you are refereing to? This is just a suggestion but I think that would help the reader | Thank you for the suggestion, I have implemented the change throughout the manuscript. |

| 21 | Line 106:
Also, I find the wording of point 1 quite confusing, it becomes clearer eventually when one reads the rest of teh paper, but rephrasing would be useful | Yes, I had difficulty properly articulating what I meant here, but the reviewer is right that it is important to get these points right. I have changed the points to:
" 1) directly influencing the stress distribution in the glaciers such that the disintegration of the land-fast sea ice caused an instantaneous speed change on the grounded ice, and 2) reducing stresses in the ice shelves which would have otherwise been too great for the ice shelves to withstand. This latter mechanism is a second order effect of buttressing, the implication being that the disintegration of the landfast sea ice in turn caused the disintegration of the ice shelves via loss of buttressing, and hence the loss of the ice shelves as a control on the upstream dynamics."
I hope these make more sense! |
|----|----|----|
| 22 | Line 106:
I don't think this distinction makes much sense. The stresses in the land ice are coupled, both grounded and floating, and so a change of stress at the ice front translates to stress changes within the ice everywhere. You can then investigate all sorts of transient effects, but I don't think that is what is done here | You are right, the redistribution of stress as boundary conditions change happens across the whole glacier, not separately in the grounded and floating ice. This is the "direct buttressing" being referred to in point #1. The "indirect buttressing" of point #2 does indeed refer to a transient response in which buttressing were responsible for ice shelf stability. After that buttressing was taken away, this could have caused the ice shelves to disintegrate which, in turn, caused the grounded ice to speed up. Evidently, we cannot simulate this fully as we don't have an appropriate calving law, but we investigate whether the first part is plausible. We find that it is implausible, suggesting an alternative mechanism like increased ocean swell or buttressing from mélange.
I hope the revised wording of points #1 and #2 make the distinction clearer. |
| 23 | Line 109 "This latter possibility is to be contrasted with other non-buttressing effects...":
where, how? | In the mind of the reader. |
| 24 | Line 110 "... capacity of sea ice to bond fragments of mélange together, prevent small calving events at the glacier terminus and the export of icebergs":
how is this influencing ice tongue stability if not through buttressing? | The 'buttressing' we refer to in analogy with ice shelf buttressing does not include these things. Instead, it refers to a change in viscous stress within the glacier due to the impact of sea ice on the stress boundary condition at the calving front. |
| 25 | Line 124 "At this point we do not include sea ice in the model geometry, so the glaciers terminate in open sea.":
but the sea ice was there, so why does it make sense to assimilate to velocities observed in the presence of land fast sea ice, but not to include its effect (as a thin ice shelf as done below) when performing the inversion? | Good point. Please refer to the answer given to this question in the reviewer's general comments above. |

| 26 | Line 138 "Fig. S5":
 I don't see how the very patchy freeboard measurements inform about the smoothness of the stresses within the sea ice. Please elaborate on that. | Thank you for pointing out this mistake. The reference was supposed to be to Fig. S4 - the deformation field over the sea ice. I have changed this in the revised manuscript. |
|---|---|---|
| 27 | Line 141 "... relative abundances of meteoric and congelation ice...":
 are you talking about land ice now? | By "meteoric" I mean ice that originates from precipitation, rather than land ice. |
| 28 | Line 148 "... such an 'ice shelf' ...":
 maybe "modeled ice shelf"? | I have changed this to "...such landfast sea ice...". |
| 29 | Line 150 "... sea ice-like ice shelf ...":
 just pick a simple term for the modeled ice shelf and keep using the same term | Apologies for the confusing mixture of terminology. I have changed this to "landfast sea ice". Here, and throughout the rest of the modelling sections, context should make it clear whether we are referring to modelled or observed landfast sea ice. |
| 30 | Line 151 "... we use a regularised-Coulomb sliding law...":
 is that the same as sliding law used for the inversion? If not, please, justify | It was the same sliding law as used for the inversion, yes. I have not added anything to the manuscript because it doesn't really matter which sliding law is used in the inversion vs forward simulation as long as the basal stresses match in the two cases. |
| 31 | Line 160 "Again, this might lead to stiffer ice than in reality...":
 what do you mean by again? I think the key here is that still you are modeling a stronger buttressing effect than the sea ice can give, which is fine, given the conclusions of its inability to provide sufficient buttressing | I have removed the "Again" as it was a bit confusing. You are right about our ice providing greater buttressing. |
| 32 | Line 164 "...change smoothly...":
 do you mean smoothly as a function of sea ice thickness? | Yes. I have added "...as a function of landfast sea ice thickness..." to the sentence to make this clearer to the reader. |
| 33 | Line 169 "This is to be contrasted with the much larger $2-5\%$ changes in speed seen at the calving fronts and on the floating ice tongues in the simulations with sea ice thickness of 10 m (Fig. 2 b, c).":
 I don't understand this sentence. How is 2-5% much larger than 15-50%? | I meant to be referring to the change of $0-1\%$ of the grounded ice speed being smaller than the 2-5% on the floating ice. I have changed the sentence to read: "These changes in grounded ice speed are to be contrasted..." |
| 34 | Line 185 "now see that the sensitivity of grounded ice speed to changes in effective sea ice thickness is dwarfed in comparison to changes in the effective thickness of glacier ice.":
 I don't understand the last part of this sentence | I apologise for the confusion. I have replaced this with the sentence:
 "we now see that the sensitivity of grounded ice speed to changes in the effective thickness of landfast sea ice is minute in comparison to its sensitivity to changes in the effective thickness of glacier ice." |
| 34 | Figure 2:
 I m not sure if the panel c is necessary, it really doesn't show anything new that panel b wouldn't have already | This is a good point. These percentage changes in speed are not part of the revised Figure 2. |

| 35 | Figure 2 caption:
Please add the glacier names on the plot | I have implemented this suggestion. |
|---|---|---|
| 35 | Line 189 "...the parts of the glaciers that showed speed increase in 2022...":
it would be helpful to have in figure 3b a comparison figure to where along these profiles acceleration was observed. Spatial profile of acceleration is not clear from any figure in sufficient detail. | This is a good suggestion, and has been implemented in the revised figure 2. |
| 35 | Figure 3 caption "...artificial landfast sea ice...":
what do you mean by artificial? modeled? prescribed? | I have removed the word 'artificial' as it wasn't adding anything. Though all of the reviewer's suggested alternatives are correct. |
| 36 | Figure 3 caption "...transects of data were collected...":
specify modeled data (not to confuse with observed) | Good point. I have changed this to "...transects of modelled speeds were collected..." |
| 37 | Figure 3 caption - referring to glacier names:
put these names on the plot, please | I have implemented this suggestion. |
| 38 | Line 203 "...thinner sections of ice...":
sea ice of glacier ice? Please specify averywhere throughout the paper what you mean by ice when | I have added the word "glacier". |
| 39 | Figure 4 caption "...10m thick sea ice.":
Is this linear? That is, does the addition of each additional 10 m produce comparable change in stresses? The formulation in the legend suggests that. | I am not sure what the reviewer is referring to here. It is the case that the speed changes are roughly linear with sea ice thickness, and probably stress too. |
| 40 | Line 216 "We cannot rule out the hypothesis that such processes were involved in the latter calving events on Crane Glacier and those on HGE starting in September 2022, though it seems likely that the effect would have been more due to the preceding loss of seaward sections of the ice tongues as opposed to this initial loss of sea ice buttressing.":
I got a bit lost now. Are you trying to explain velocity changes on grounded ice? Why not repeat the same excercise, remove the floating ice tongues in the model and see what effect it has on the grounded ice in terms of instantaneous speedup? | I apologise, this is a confusing sentence. I am not trying to explain velocity changes on the grounded ice (in general, I don't have much interest in that). I am trying to explain changes to the calving behaviour of the ice shelves. I have changed this to read:
"We cannot rule out the hypothesis that such processes were in part responsible for the elevated calving rate on Crane and HGE Ice Shelves starting in September 2022. However, if this were the case, it seems likely that the calving events themselves would have had a greater impact on subsequent calving rate than the loss of sea ice." which I hope is clearer. |
| 41 | Line 218 "... the effect ...":
The effect of what? | Thank you for pointing out this confusion, this sentence was not well written. See my response to comment #40. |
| 42 | Figure 5 c:
Is this T2m change or anomaly? make label consistent | I have changed the label on the figure to read 'anomaly'. |

| 43 | Figure 5:
Please mark the date of fast ice breakup on plots a b and d. Also, if your timseries goes back enough, also mark the date of the 2002 ice shelf breakup | I have added vertical lines to the plots showing the dates of these events as suggested. |
|---|---|---|
| 44 | Line 262 "Observations of ice shelf basal melt rates":
estimates | I have made the change as suggested. |
| 45 | Line 265 "A timeseries of the mean basal melt rate from this region shows":
Also, it would be worth mentioning that these basal meltrates have not been validated, and if anything, similar estimates by Adusumilli at al have been shown to produce variability and order of magnitude larger than observed in situ on the neighboring Filchner-Ronne Ice Shelf. | Would it be possible to provide a reference for this? |
| 46 | Line 266 "0 ma$^{-1}$":
I think you need to mention the giant $\pm 4$ m a$^{-1}$ uncertainty here | Very good point! I have added the uncertainty. |
| 47 | Line 351 "This leads us to suggest that the term "buttressing" should not be used in the context of sea ice in the way it is understood when applied to ice shelves.":
I think I know what you mean here but it could be said in a less ambiguous way. | I think this is quite a precise statement. I don't want to say "People shouldn't use the term 'sea ice buttressing' ", but I do want to say "People shouldn't use the term 'sea ice buttressing' in a way analogous to ice shelf buttressing." Perhaps that would be better? |
| 48 | Figure S3:
Figure S3: What is the vertical orange line on the western side of scar inlet? Did the calving front advance so much in less than a year from that vertical orange to the horizontal purple line further northwest? | Good spot! A new figure with updated calving fronts has replaced the old one. |

**References**

[revised manuscript text omitted]

---

## Author Comment (AC4)

**Responses to community comment #1 for the article "The impact of landfast sea ice buttressing in the Larsen-B Embayment, Antarctica"**

We would like to thank very much N. Ochwat and T. Scambos for taking the time to read this article, and for initiating a constructive and open discussion about this subject of shared interest. We agree entirely with the comments they have raised, and have implemented/will implement revisions that address some of them. Where this is not so, it is in general where the work would deviate too greatly from the main focus of the article - often by going into further detail about what precisely caused the changes in calving behaviour and dynamics of the glaciers, while the intention of the article is to study singularly whether landfast sea ice buttressing (defined in analogy to ice shelf buttressing) could have been responsible for these changes. We hope that the revised manuscript clarifies certain points of confusion collectively highlighted by the reviews, and that the discussion more completely describes the conclusions and limits of the study.

A number of interesting points are raised in the comments covering a range of features of our article. We respond individually to the points raised below. Comments are shown in violet and our responses in black. The first three comments relate to specific choices regarding the geometric setup of the model so we have aggregated them.

- Regarding the methods section and the data used in the modelling, we agree with the reviewer 1 that more information is needed about the methods and data used. In particular, selected bedrock data as well as mapped location of the grounding zone may play an important role in the model's results. There are two primary options for bedrock data, Huss and Farinotti (2014) and BedMachine (Morlighem et al., 2023). Bedmachine data has recently been updated to incorporate sonar derived bedrock information for the terminus area of Crane, collected from the RV N.B. Palmer in 2006. Additionally, the Huss and Farinotti (2014) data do not cover the whole study area displayed in Fig. 1. What did the authors do in these regions? Regarding the grounding line, it is unclear how this was determined and why it was chosen for the model; it would also impact how the glacier acceleration is discussed (i.e., floating or grounded ice). We note that the paper cited in support of the grounding line is not available at this time for review. It would be interesting to discuss the effect of using other, published, grounding lines (e.g. Rott et al., 2018, Sun et al. 2023, Ochwat et al. 2023).

  Furthermore, the model output of the "glacier terminus" (cyan lines figure 1) is likely incorrect as there is 300+ m thick ice several km further downstream from the glacier terminus identified in the model/figure 1. How are the authors defining the glacier terminus?

  What are the potential problems incurred by using a 2015 DEM with a 2021 Grounding line?

  Some similar queries were raised by the set of solicited reviewer comments. We have implemented revisions that aim to clarify the methods in general. Firstly, to address the calving fronts in the model, we realise we could have done a better job

in making the ice shelves look more like they did in reality in 2021. For the revised manuscript, we created a new model geometry with larger ice shelves and performed the initialisation and each modelling experiment again. This has not changed the results of the article, but they are perhaps more convincing now the glaciers look more accurately rendered.

The bedrock topography of the Antarctic Peninsula is poorly mapped and all of the available bedrock topography products have flaws. For example, in BedMachine v3, there are many fast-flowing ($> 500$ ma$^{-1}$) areas where the ice thickness is $< 20$ m, which is clearly unrealistic. The Huss and Farrinotti (H&F) (2014) dataset seemed to have fewer of these areas of implausibly thin ice. We therefore used the H&F (2014) dataset, along with the REMA 200 m DEM, to construct a continuous grid of ice surface elevation and ice thickness amenable for modelling at 125 m resolution. Some minor modifications to the H&F grid were necessary to produce continuous thicknesses and surface elevations including smoothing at the grounding line, and the removal of steps in bedrock elevation at the edge of the H&F domain.

As with the bed topography, the grounding line of the Antarctic Peninsula is poorly mapped and most grounding line products are either discontinuous or valid for times prior to the late-2000s. We use a new grounding line product, now detailed in a pre-print submitted by Wallis et al., which addresses the problems presented by discontinuous or old grounding line products.

In general, though it is possible that differences in geometry products would result in different conclusions, this does not seem likely among those that could plausibly reflect the real glacier geometry. This is because the arguments we make in the paper are simple and mechanical. We have shown the results to be insensitive to different calving front geometries having now carried out the experiments for larger ice shelves. Though it would be nice to carry out a greater number of experiments showing more completely the insensitivity of the results to changes in geometry, it would take a fairly large amount of effort that we do not feel is warranted.

- In Figure 5, please include more years as this will help with understanding the timing of the events. We also request discussion on the fact that melt was lower in 2022 than previous years, according to AMSR data and that the largest peak in T2M was in 2021.

The section of the article discussing additional environmental forcing at the time of the sea ice evacuation (promoted out of the discussion in the revised version) is not supposed to be a focus of the article. Instead, we hoped to include it as something of additional interest noting the complexities of the environmental factors that could have been in play at the beginning of 2022. Though it would be nice to extend this section, it is not necessary for the conclusions of the article so we do not intend to add additional dates to extend the timeseries. However we will include reference to the AMSR melt estimates as suggested.

- The connection between basal melting and the near instantaneous break-up of the ice tongues after fast ice removal is unclear. Can the authors discuss how those two processes are connected? Also, do the authors have observation or model (reanalysis) indications of a large swell event leading to the breakup of the ice

tongues? The relationship between the basal melting, swell, and ice tongue response is generally unclear.

It is true that this section regarding basal melting was a bit unclear. We did not present reanalysis or observations of swell around the time of the sea ice breakup. This information is already presented in your own pre-print on the event, which we will cite at this point in the manuscript, and we did not feel that repeating the analysis would strengthen our conclusions regarding the impact of landfast ice buttressing. We did choose to present the basal melt data because these are novel observations that open avenues towards alternative interpretations regarding the cause of the ice tongue breakup and glacier speed-up, which we feel enriches the discussion in the manuscript. Indeed, given the large magnitude of the basal melt anomaly, we feel it would be remiss to exclude those observations from the discussion. Again, these suggestions are not intended to be particularly conclusive, but we hope the revisions made to the article make the parameters of the discussion more well-defined.

- We also would like to see the results discussed in the context of Robel, 2017.

Robel, A. Thinning sea ice weakens buttressing force of iceberg mélange and promotes calving. Nat Commun 8, 14596 (2017). https://doi.org/10.1038/ncomms14596

This has been taken into consideration in the revised manuscript, which includes additional discussion of the potential coupling that iceberg mélange can provide between landfast sea ice and ice shelves.

Specifically, the revised manuscript goes into a bit more detail about the mélange in the proglacial embayments in front of HGE and Crane Glaciers (that is not easily distinguished from the ice shelves themselves). It is quite possible that these transition regions, where the ice shelf becomes mélange that rarefies until it is entirely sea ice, (not dissimilar from those considered in Robel 2017) enable dynamic coupling between the landfast sea ice and the glaciers as they might be both able to buttress the ice shelves, and be sensitive to the presence of landfast sea ice. We do not model these regions of mélange as the study is interested in the buttressing of the landfast sea ice, but it is discussed as a point of future work. Specifically, we acknowledge a potential second-order impact of sea ice buttressing that we could not investigate where loss of buttressing causes a disaggregation of mélange which destabilises the ice shelves. However it should also not be assumed that mélange broke up because of loss of buttressing. For example, the mélange is likely to be susceptible to the same forcing that caused the sea ice disintegration.

- The paragraph in Lines 326-336 is not clear.

Yes, this is just supposed to suggest that that the buttressing effect of landfast sea ice and its other influences might vary differently as a function of landfast sea ice extent. E.g. the buttressing effect of fast ice confined to the proglacial embayments is likely to be as large as if fast ice covered the whole Larsen-B Embayment, but its ability to reduce swell might be reduced. This could be a way of differentiating the influence of these effects. We will try and make this clearer in the revised manuscript.

- Lastly, the connection between the ice tongues' disaggregation and the loss of the fast ice needs to be better justified if the fast ice was not buttressing the tongues. The authors are implying an alternative mechanism for the instantaneous response if the fast ice was not offering any sort of buttressing effect, but the path to disaggregation is not clear in that case.

The mechanism alluded to above involving iceberg mélange is a potential candidate cause of the ice shelf disaggregation, though even in its revised form the article does not take a firm position on precisely what caused the calving immediately after the sea ice evacuation. We do not consider this to be a particularly big issue as there is no reason to believe that loss of sea-ice buttressing is the only mechanism by which ice shelf disaggregation should occur with the loss of the landfast sea ice. However, we have taken the lack of clarity of this point into consideration in the revised manuscript.

---

## Referee Report (RR1)

**GENERAL COMMENTS**

The authors have made significant improvements to the paper in response to my and the other reviewers' comments. I especially appreciate the revisions to Section 2 and the new section that discusses the use of a viscous flow law for sea ice.

One point, which I don't want belabor too much, is that in my review I suggested that the sea ice buttressing force depends to first order on the sea ice thickness (to a power of 2) and that extension/compression will decrease/increase the buttressing force. The authors response implies that Larsen-B system is more complicated because of its 2D geometry and that the equations I wrote down don't account for shear stresses that would arise in that situation. This is actually not the case. The equations I wrote down were really just statements about the stresses at the glacier-sea ice boundary. The shear stresses will affect both the local sea ice thickness at the glacier-sea ice boundary as well as the tectonic/deviatoric stresses there. I don't expect the authors to address this point anymore than they already have, I just wanted to clarify the point that I was trying to make.

I still struggle a little with the terminology in the paper, which I think could be used more precisely. The authors argue that sea ice buttressing does not affect glacier flow, yet they show that (i) the ice shelves can speed up if the sea ice is removed and (ii) ice shelf retreat leads to speed up of grounded ice. Ice shelves are just extensions of the grounded ice, so isn't the sea ice therefore buttressing the glaciers? Stress perturbations at glacier termini can decay pretty rapidly, and so maybe it shouldn't be that surprising that velocities 20 km from the ice shelf edge don't change very much when sea ice is removed.

I agree with the authors that sea ice doesn't have an immediate impact on ice flow far upstream; I'm just not sure if it is correct to say that the sea ice is not buttressing the glaciers. Related to this, I think the authors could put more emphasis on the timing of events; there is a pretty long lag between break up of sea ice and acceleration of the grounded ice, suggesting a sequence of events in which sea ice distintegration leads to ice shelf break leads to acceleration of grounded ice.

**SPECIFIC COMMENTS**

L9-12: This is a little confusing. Are you saying that the loss of sea ice buttressing caused the ice shelves in front of these four glaciers to speed up but not the grounded ice? Or some glaciers but not others? I think part of the confusion stems from the fact that the paper talks about all of the glaciers in the Larsen-B Embayment but the abstract specifically discusses just four of them.

L53: This is somewhat ambiguous. Were there no calving events from 2001–2022? Was it just in 2022 that it was difficult to define the calving fronts?

L81: I'm not convinced from Fig. 1b that Crane Glacier started to accelerate in February 2022, especially when you consider the uncertainty shown in the plot.

Fig. 1a: The color (white) for the 2021 sea ice extent is not visible in the legend because of the white background. Perhaps consider adjusting the color scale so that the last line isn't white?

L96: Cite Fig. S3b?

L228–230: I agree that the landfast sea ice removal did not cause a large, instantaneous change in speed at/above the grounding line, but I'm not sure if you can say it wasn't the primary cause. Would the glaciers have sped up if the sea ice had remained intact?

L347: But it does seem to have a buttressing effect on the ice shelves, which are part of the glaciers.

L441–442: Maybe I missed it, but I don't see how the results from this study support this claim.

---

## Author Response (AR2)

**Responses to second set of reviewer comments for the article "The effect of landfast sea ice buttressing in the Larsen-B Embayment, Antarctica"**

We want to thank again the reviewers and editor for the time they have spent evaluating this article and for providing such valuable feedback.

Please find my responses to the final comments of reviewers #1 and #3 below. As before, my responses to general comments are in teal, while specific comments are tabulated afterwards.

Changes are confined largely to small edits in the wording of the abstract, and changes to section 4 "Environmental drivers". There are no significant edits to the figures.

**Responses to comments from anonymous Reviewer #1**

**Reviewer 1:** I think this paper is really great and the authors have exhaustively answered all my questions. One exception to that is section 3.3 - Environmental drivers.

We thank the reviewer very much for their praise of the revised manuscript. We agree that the work invested during this review process has improved the article - and thank them for their significant part in that.

As the authors admit 3.3 is more of a detour, but unlike the rest of the paper, which is really clear, well-reasoned and to the point, this section is highly speculative and doesn't do justice to the rest of the paper. I think it would be best to leave it out.

While the reviewer is still not a fan of section 4 (which was section 3.3 prior to my edits following the first set of reviewer comments) on environmental drivers, my co authors and I have decided to keep it in the paper, with various small changes, for the reasons I set out below. Firstly, though section 4 contains some conjecture, I don't think it's fair to call the whole section 'highly speculative', rather it presents a set of quantitative observations and reanalysis data that are used to evaluate the wider environmental change at the time of the sea ice breakup. Reanalysis data on temperatures and wind speeds are complemented by satellite observations of basal melt rates on the one remaining section of ice shelf (Scar Inlet) which is large and intact enough for this dataset to be produced. This information has not been included in other publications discussing the Larsen-B sea ice breakup, so we feel it is an interesting contribution to the literature. We would have liked to also include some ocean temperature observations as the reviewer suggests (see their comment below), as this may have strengthened our suggestion that higher ocean temperatures caused the larger basal melt. We had conversations with our oceanography colleagues about sourcing this, however, as the timing of the sea ice breakup coincided with the COVID19 pandemic there is simply no in situ temperature data from this time

as ocean cruises were not able to take place.

In terms of edits, I have removed a couple of sentences that speculate on the contribution of warm water to mélange disaggregation and increased calving rate. Additionally, I have made statements regarding the timing of changes in basal melt rates over Scar Inlet less precise to reflect the large uncertainties in the estimates.

Ultimately, though some of the results in the section could be given more attention in a separate piece of work, I don't think leaving the section with these edits does any harm to the article, and doing so allows us to present data that some readers will find interesting.

| Reviewer 2 | | |
|---|---|---|
| ID | Reviewer Comment | Response |
| 1 | There no details about how these melt rates were inferred and how they were validated. The uncertainties are so high, that a straight line could be fit through the data over almost entire duration of the record. The time series is from the grounding line region where bending effects are typically present - was this glacier dynamics taken into account or is hydrostatic approximation used in that region as typically done? | We provided the citation to Gourmelen et al. (2017) which explains in detail how the basal melt rate data is produced and validates this method over an ice shelf in the Amundsen Sea Embayment. We have added the a to Davison et al. (2023) which presents the same basal melt rate data (along with other parameters) across the whole of Antarctica. I disagree slightly with the reviewer that it is meaningful that that a straight line could almost be drawn through the basal melt data presented in Figure 5d. If one were to analyse how the data shown here were evidence for different timeseries models, even given the error, the likelihood associated with a linear model would surely be small, especially when you consider that we are looking at a large number of densely spaced datapoints. Regardless, I have changed the wording of the sentence describing the timeseries to reflect that, due to the large uncertainties, the timings of specific changes in melt-rate should not be stated too precisely.

 The reviewer is right to say that the hydrostatic condition is used to infer basal melt rates, but the timeseries is not taken from the region around the grounding line, where this condition might not hold. |
| 2 | What is the ocean structure here and is it plausible increased ocean temperatures (in presumably the lower part of the water column) actually reach the melange which is much thinner than the grounding line location? | I have removed sentences referring to the interaction of warm water with the mélange or calving fronts. (Also, see comment above regarding the lack of in-situ measurements of ocean structure.) |

| 3 | If melt rates have been increasing for some time already, was there any noticeable variability in the glacier motion that could be attributed to changing melt rates? | This is a very good question! The timeseries data at the grounded locations we analysed do not show significant dynamic changes before 2022 compared to the dramatic changes we see afterwards. However, further work is required to establish whether there is a signal detectable anywhere in either dynamics or grounding line location. I have added the following line stating this point in the article: "Further work is required to establish whether the deeply grounded glaciers of the Larsen-B Embayment, and perhaps beyond, exhibit a dynamic signal before the landfast sea ice evacuation that could point to the influence of enhanced ablation due to ocean warming.". |
|---|---|---|
| 4 | If anything, what seems more plausible is that enhanced subglacial runoff would be responsible for such increase in melting than sudden increase in ocean temperature (within the values that are realistic). | This is a good thought and one we have considered, as subglacial melt and associated ocean plumes can enhance turbulent mixing and melt at the grounding line/ice shelf base. However we don't have any evidence to show that this was occurring in the study region at the time of the change. Future studies could investigate the runoff parameter in regional climate models to assess whether the runoff is likely to be higher than the usual seasonally driven surface melt that occurs in this region. The increase in basal melt rates we measure appears to extend over a longer multi-year period with only subtle seasonality, so this might suggest that another process such as ocean warming has more of an effect. |
| 5 | What is the biyearly variability in the melt rates - is that a filtering artifact or is there any explanation for melt rates varying on a 2-yearly timescale? 2 years is a bit odd yet there seems to be a signal of that period in the inferred melt rate. | We are not sure what is behind this ∼biennial variation in the data as, like the reviewer suggests, the 2-3 year period is not immediately indicative of any processes that come to mind. It might well be an artefact of the data. However, the signals of interest occur on a different timescale with larger amplitude, so these do not influence the statements made in the section. |
| 6 | How does increased melt in the grounding zone cause rapid calving? | On Crane glacier, after the initial disaggregation of ice mélange and the most seaward part of the calving front, the calving front was likely to be close to the grounding zone - a configuration in which warm water could, for example, induce a calving multiplier effect. However, I have removed mention of the impact of warmer water on calving as the mechanism is different for the two ice shelves of interest and dependent on factors we do not consider in the data. |

**Responses to comments from Reviewer #3**

| | Reviewer 2 | |
|---|---|---|
| ID | Reviewer Comment | Response |
| 1 | L9-12: This is a little confusing. Are you saying that the loss of sea ice buttressing caused the ice shelves in front of these four glaciers to speed up but not the grounded ice? Or some glaciers but not others? I think part of the confusion stems from the fact that the paper talks about all of the glaciers in the Larsen-B Embayment but the abstract specifically discusses just four of them. | I have changed the abstract in a couple of places to help disambiguate whether the statements refer to the four glaciers the study is concerned with or all the glaciers in the region. I have changed the particular sentence in question to:
 "The results show that direct landfast sea ice buttressing had a negligible impact on the dynamics of the grounded ice streams. Furthermore, we suggest that the loss of landfast sea ice buttressing could have impacted the dynamics of the rheologically weak ice shelves, in turn diminishing their stability over time, however, the accompanying shifts in the distributions of resistive stress within the ice shelves would have been minor. This indicates that this loss of buttressing by landfast sea ice is likely to have been a secondary process in the ice shelf disaggregation compared to, for example, increased ocean swell or the drivers of the initial landfast sea ice disintegration." |
| 2 | L53: This is somewhat ambiguous. Were there no calving events from 2001–2022? Was it just in 2022 that it was difficult to define the calving fronts? | I have changed this to read: "For much of the period prior to the first calving events of 2022 the transition between consolidated ice shelf to landfast sea ice appeared smooth in satellite images, encompassing a region of ice mélange, making the calving fronts difficult to define precisely." . |
| 3 | L81: I'm not convinced from Fig. 1b that Crane Glacier started to accelerate in February 2022, especially when you consider the uncertainty shown in the plot. | I have changed this to 'early-to-mid 2022'. |
| 4 | Fig. 1a: The color (white) for the 2021 sea ice extent is not visible in the legend because of the white background. Perhaps consider adjusting the color scale so that the last line isn't white? | I have changed the background of the legend to grey so each line is visible. Thank you for pointing this out. |
| 5 | L96: Cite Fig. S3b? | I have added this reference to figure S3 b (alongside a reference to figure 1 f). |

| 6 | L228–230: I agree that the landfast sea ice removal did not cause a large, instantaneous change in speed at/above the grounding line, but I'm not sure if you can say it wasn't the primary cause. Would the glaciers have sped up if the sea ice had remained intact? | I have changed this line to read:
"This is an indication in its own right that the buttressing effect of landfast sea ice was not its primary control on the dynamics of the glaciers of the Larsen B Embayment."
The reviewer is quite right, the wording here is not correct and I think the glaciers would not have sped up had the sea ice remained intact. |
|---|---|---|
| 7 | L347: But it does seem to have a buttressing effect on the ice shelves, which are part of the glaciers. | I have changed "landfast sea ice does not have the ability to buttress glaciers" to "landfast sea ice has limited ability to buttress glaciers". Thank you for pointing out this inconsistency. |
| 8 | L441–442: Maybe I missed it, but I don't see how the results from this study support this claim. | We show it is likely that the perturbation to the viscous stress in the ice shelves with the removal of the landfast sea ice buttressing was small compared to the spread of viscous stress within the ice shelves. Hence, a widespread disintegration of the ice shelves is not likely to have been caused by this change. |

**References**

Davison, B. J., Hogg, A. E., Gourmelen, N., Jakob, L., Wuite, J., Nagler, T., Greene, C. A., Andreasen, J., and Engdahl, M. E.: Annual mass budget of Antarctic ice shelves from 1997 to 2021, Science Advances, 9, eadi0186, https://doi.org/10.1126/sciadv.adi0186, 2023.

Gourmelen, N., Goldberg, D. N., Snow, K., Henley, S. F., Bingham, R. G., Kimura, S., Hogg, A. E., Shepherd, A., Mouginot, J., Lenaerts, J. T. M., Ligtenberg, S. R. M., and van de Berg, W. J.: Channelized Melting Drives Thinning Under a Rapidly Melting Antarctic Ice Shelf, Geophysical Research Letters, 44, 9796–9804, https://doi.org/10.1002/2017GL074929, 2017.